# Alternative sulphur metabolism in the fungal pathogen *Candida parapsilosis*

Lisa Lombardi [1] ✉, Letal I. Salzberg[2], Eoin Ó Cinnéide [1], Caoimhe O'Brien[1], Florent Morio [3], Siobhán A. Turner[1], Kevin P. Byrne [2] & Geraldine Butler [1] ✉

*Candida parapsilosis* is an opportunistic fungal pathogen commonly isolated from the environment and associated with nosocomial infection outbreaks worldwide. We describe here the construction of a large collection of gene disruptions, greatly increasing the molecular tools available for probing gene function in *C. parapsilosis*. We use these to identify transcription factors associated with multiple metabolic pathways, and in particular to dissect the network regulating the assimilation of sulphur. We find that, unlike in other yeasts and filamentous fungi, the transcription factor Met4 is not the main regulator of methionine synthesis. In *C. parapsilosis*, assimilation of inorganic sulphur (sulphate) and synthesis of cysteine and methionine is regulated by Met28, a paralog of Met4, whereas Met4 regulates expression of a wide array of transporters and enzymes involved in the assimilation of organosulfur compounds. Analysis of transcription factor binding sites suggests that Met4 is recruited by the DNA-binding protein Met32, and Met28 is recruited by Cbf1. Despite having different target genes, Met4 and Met28 have partial functional overlap, possibly because Met4 can contribute to assimilation of inorganic sulphur in the absence of Met28.

Fungal pathogens are a major cause of morbidity in immunocompromised patients, and invasive fungal infections kill around 1.5 million people every year[1]. Since fungi are eukaryotes, the development of new antifungals without side effects is challenging and slow-paced[2].

Among fungal pathogens, *Candida* spp commonly causes invasive infections in Intensive Care Units (ICUs) worldwide, with mortality rates >40%[3,4]. Most well-studied *Candida* species belong to the monophyletic CUG-Ser1 clade, where the codon CUG is translated as serine rather than leucine[5]. Although *Candida albicans* is the most prevalent, invasive infections caused by non-*albicans* species have steadily risen during the last 30 years[3]. Among them, *Candida parapsilosis* is the 2nd/3rd most frequently isolated species in the ICU[3].

*C. parapsilosis* is frequently found as a commensal in the human gut[6], but it is also ubiquitous in the environment[7,8]. This opportunistic pathogen easily transmits horizontally (i.e., from healthcare workers to patients), and it has been associated with nosocomial infection outbreaks worldwide, many of which are caused by fluconazole-resistant isolates[9,10]. *C. parapsilosis* disproportionally affects patients undergoing parenteral nutrition and premature infants[3,11]. Moreover, antimicrobial prophylaxis in patients undergoing allogeneic hematopoietic cell transplant is associated with the clonal expansion of *C. parapsilosis* in the gut at the expense of the resident bacterial flora[12,13]. This is associated with worse overall survival and higher transplant-related mortality, in some cases also leading to translocation of *C. parapsilosis* into the bloodstream[12,13].

The molecular function is generally conserved between orthologous genes of closely related fungal species, thus facilitating the study of the virulence of related pathogens. Nonetheless, rewiring of transcriptional regulatory networks has been observed between *Candida* species and *Saccharomyces*[14–16], and within *Candida* species[17–20]. Some examples include Cph3, which regulates hyphal growth in *C. albicans* and hypoxic response in *C. parapsilosis*, and several regulators of

[1]School of Biomolecular and Biomedical Science, Conway Institute, University College Dublin, Belfield, Dublin, Ireland. [2]School of Medicine, Conway Institute, University College Dublin, Belfield, Dublin, Ireland. [3]Nantes Université, CHU Nantes, Cibles et Médicaments des Infections et de l'Immunité, UR1155, Nantes, France. ✉e-mail: lisa.lombardi@ucd.ie; gbutler@ucd.ie

biofilm formation that play a role in *C. albicans* only (*BRG1* and *TEC1*) or *C. parapsilosis* only (*CZF1*, *UME6*, *CPH2* and *GZF3*)[19]. As a result, despite their phylogenetic relatedness, the pathobiology of *C. parapsilosis* cannot always be directly extrapolated from *C. albicans*[3,19], and studying *C. parapsilosis* is pivotal to understanding its specific virulence determinants.

The ability of *C. parapsilosis* to thrive in environments as different as the human gut, soil, and the rubber seal of washing machines suggests that this opportunistic pathogen may have significant metabolic flexibility[7,8,21]. The ability to finely tune the metabolism to adapt to different niches is a well-known virulence factor of fungal pathogens, and understanding how fungi use different sources of carbon, nitrogen, and trace metals is informing drug discovery[22,23]. Sulphur metabolism is of particular interest, as it is not conserved between host and fungal cells, making it a promising drug target[24].

Sulphur is found in cysteine and methionine, and many essential molecules like coenzyme-A, glutathione (essential for maintaining the intracellular reducing environment), S-adenosylmethionine (SAM, the main methyl donor in the cell and a precursor for the synthesis of phospholipids, vitamins, polyamines), and iron-sulphur clusters. The pathway leading to the synthesis of cysteine and methionine is well characterised in the model yeast *Saccharomyces cerevisiae* (reviewed in ref. 25). Inorganic sulphur (sulphate, $SO_4^{2-}$) is imported into the cells through sulphate permeases, activated to adenylyl sulphate (APS) and then to 3′-phospho-5′-adenylyl sulphate (PAPS), and further gradually reduced by NADPH to sulphite ($SO_3^-$) and then sulphide ($S_2^-$). Sulphide is then incorporated into homocysteine. Homocysteine lies at the crossroads between the transsulfuration pathway, which results in the production of cysteine (and, further on, glutathione), and the methyl cycle, which leads to methionine and SAM synthesis.

The expression of most of the yeast genes required for sulphur amino acid biosynthesis relies on the transcriptional regulator Met4[26,27]. Notably, a large insertion (D504-Q599) into the C-terminal basic leucine zipper (bZIP) domain of Met4 in the *Kluyveromyces* clade and *Saccharomyces* species abolished its ability to directly bind DNA[28]. Met31/32 (two paralogous zinc finger transcription factors) and Cbf1 (a basic helix-loop-helix protein) are required to recruit Met4 to promoters in these species[27–29]. Interaction with Met28, a basic leucine zipper protein that lacks direct DNA binding activity in *S. cerevisiae*, is also required to stabilise the DNA-bound Met4 complexes[30,31].

Transcriptional regulation of cysteine and methionine synthesis in other ascomycete species is usually regulated by Met4 orthologs, but nuances in the complexity of the network have been observed[28,32]. Whereas in *S. cerevisiae* and its close relatives Met4 works in conjunction with several different cofactors, in the CUG-Ser1 member *C. albicans* Met4 contains a canonical bZIP domain and seems to require only Cbf1 to control the synthesis of sulphur-containing amino acids; Met28, on the other hand, is essential and does not appear to be involved in sulphur metabolism[32]. However, it is still not clear whether Cbf1 binds directly to Met4 or to DNA[32]. The circuitry is even simpler in filamentous fungi like *Aspergillus nidulans* and *Neurospora crassa*, in which Met4 orthologs (MetR and Cys3, respectively) retain the ability to bind DNA and act as the main (and possibly the sole) regulators controlling the pathway[33,34]. In all these species, Met4 is absolutely required for methionine biosynthesis and its disruption results in methionine auxotrophy[25,32–34]. Nothing is known about sulphur assimilation and cysteine/methionine biosynthesis in *C. parapsilosis*.

We report here the construction of 253 *C. parapsilosis* strains in which both alleles of target genes (mostly transcription factors and protein kinases) were either deleted or disrupted by insertion of a premature stop codon using homologous recombination or CRISPR-Cas9 in this diploid species. The mutant strains were combined with previously constructed deletion strains and screened for growth defects in >50 different conditions. The screen highlighted that in *C. parapsilosis*, unlike all other ascomycetes studied, the synthesis of

sulphur-containing amino acids requires both *MET4* and a putative paralog *MET28*. Transcriptomic analysis shows that in *C. parapsilosis*, Met4 and Met28 regulate different pathways used to import/assimilate organic and inorganic sources of sulphur and that pathways previously studied in depth only in bacteria are also active in yeasts. Met4 and Met28 play complementary roles in the regulation of inorganic and organic sulphur, which may contribute to scavenging sulphur in the variegated niches that *C. parapsilosis* can colonise. Analysis of *cis*-elements found in sulphur-responsive genes suggests that Met4 is recruited by the DNA-binding protein Met32, and Met28 is recruited by Cbf1.

## Results

### Disrupting genes in *Candida parapsilosis*

We previously described the construction of homozygous deletions of 73 transcription factors and 16 protein kinases in *C. parapsilosis*, generated by using homologous recombination to replace each allele with either *Candida dubliniensis HIS1* or *Candida maltosa LEU2*[19]. We have now used the same technique to delete 76 additional transcription factors (Supplementary Data 1). We also used a plasmid-based CRISPR-Cas9 system[35] to disrupt 177 genes encoding transcription factors (52), protein kinases (70), genes of unknown function (20), and genes located on chromosome 1 (35) (Fig. 1A, Supplementary Fig. 1 and Supplementary Data 1). Genes were disrupted by introducing 11 bases containing stop codons in all open reading frames and a unique tag (barcode) (Supplementary Fig. 1). Two potential disruption strains for each targeted gene (lineages A and B) were screened by PCR to confirm the presence of the barcode (Supplementary Fig. 1 and Supplementary Data 1).

Overall, 253 new mutant strains were generated. When combined with the strains previously generated in our laboratory by Holland et al. [19], the collection includes disruptions of 200 predicted transcription factors, 85 predicted protein kinases, and 66 genes with miscellaneous functions, most of which were generated as two independent lineages (Fig. 1A). Growth of the mutant collection was measured in >50 different conditions designed to identify different stress phenotypes (including presence of antifungal drugs, heavy metals, compounds affecting the cell wall or inducing osmotic/oxidative stress), as well as defects in the assimilation of different nitrogen sources and biosynthesis of different amino acids or adenine (Fig. 1, Supplementary Notes 1–2, Supplementary Table 1 and Supplementary Data 2, 3).

Overall, our screen implicated 22 transcriptional regulators and 11 protein kinases in response to antifungal drugs, heavy metals, osmotic, oxidative and cell wall stress (Supplementary Table 1). Thirty-nine genes (26 transcriptional regulators and 13 protein kinases) were associated with the use of alternative nitrogen sources and amino acid biosynthesis (Supplementary Table 1).

### Phenotypic effects of disruptions

Screening of the library on YPD supplemented with chemical stressors, synthetic defined media supplemented with different nitrogen sources, and synthetic complete media lacking specific classes of amino acids confirmed previously identified phenotypes, or supported predicted phenotypes based on the function of orthologous genes in *C. albicans* or *S. cerevisiae* (Fig. 1, Supplementary Notes 1, 2, Supplementary Figs. 2–4, Supplementary Table 1 and Supplementary Data 2, 3).

These include sensitivity to azole drugs in response to disrupting *UPC2* (*CPAR2_207280*)[19,36] or the ATPase encoding gene *ISW2* (*CPAR2_404600*)[37], hypersensitivity to hydrogen peroxide in *SKN7* (*CPAR2_304240*) disruptions[37], to oxidative stress and cadmium in *CAP1* (*CPAR2_405030*) disruptions[38,39], and to osmotic stress response and oxidative stress in disruptions of the *MAPK* kinase *PBS2* (*CPAR2_806570*)[40] (Fig. 1B). Deletion of the transcriptional regulators *SFU1* (*CPAR2_700810*) and *TUP1* (*CPAR2_109520*) led to increased

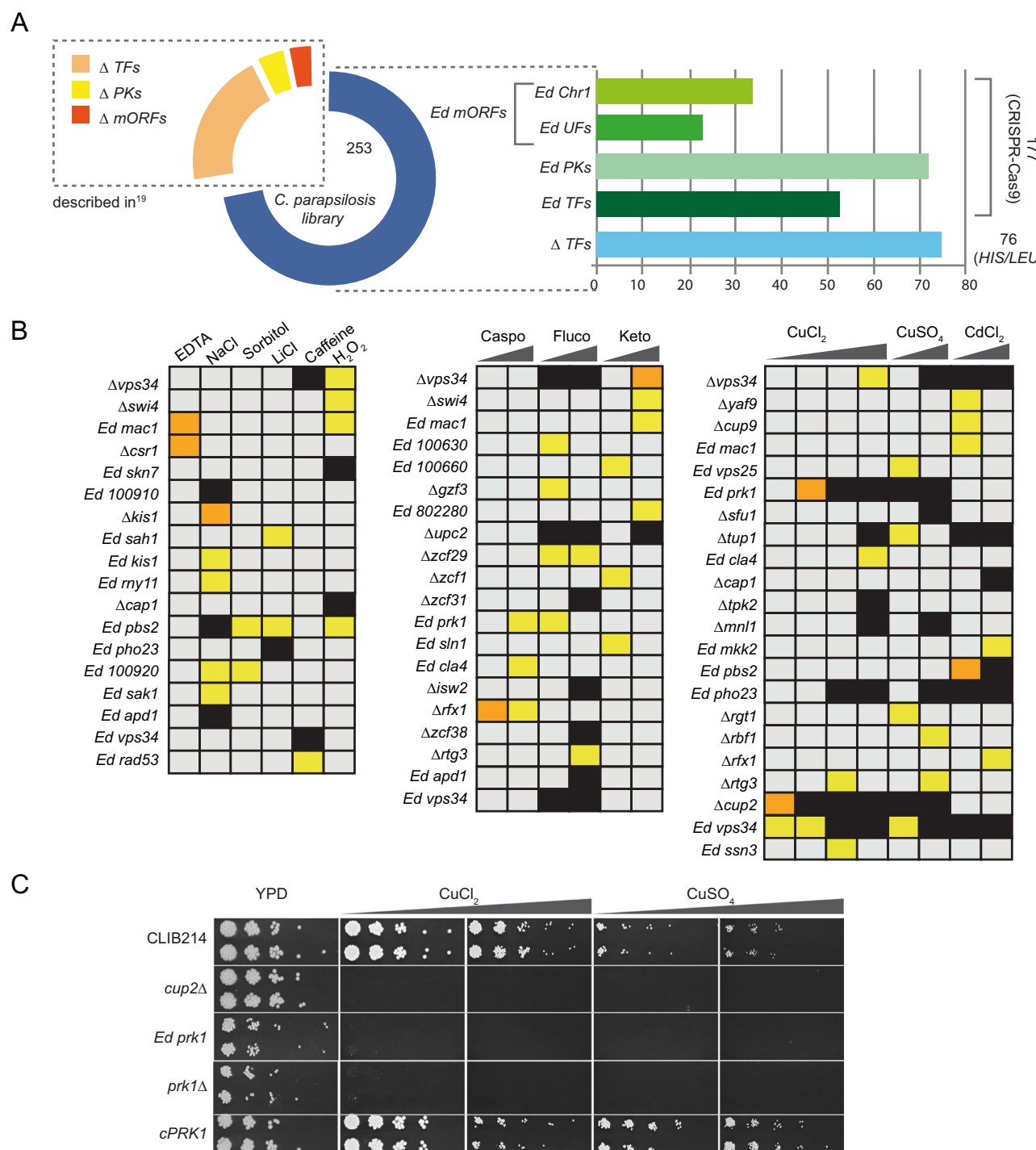

**Fig. 1 | Response of *C. parapsilosis* to oxidative and osmotic stresses, cell wall disturbing agents, antifungal drugs, and metal toxicity. A** The pie chart represents the final collection of mutants (351 strains), with the blue segment representing the new strains generated in this study. TF: transcription factors; PKs: protein kinases; mORFs: miscellaneous ORFs; UFs: unknown function. Ed = edited strain; Δ = deleted strain. **B** Growth of the mutant collection was determined in conditions designed to identify different stress phenotypes (Supplementary Data 3). The growth of each strain was compared between the control plate (YPD) and each experimental plate and strains with a Z-score > 2 or < − 2 (Ln Ratio values that were greater than two standard deviations below (−) or above (+) the mean for each plate) were considered to have a growth defect (or advantage). Strains with Z-scores between − 3 and − 6 are indicated in yellow; strains with Z-score < − 6 are

indicated in orange; complete absence of growth is indicated in black. Only mutant strains that showed a growth defect in at least one condition are included in the heatmap. The concentrations of the stressing agents are indicated in Supplementary Data 3. Caspo = caspofungin; fluco = fluconazole; keto = ketoconazole (**C**) The involvement of the protein kinase Prk1 in response to copper toxicity was confirmed in a spot assay. The growth of two independent lineages of the edited and deleted *prk1* mutants (*Ed prk1* and *Δprk1*, respectively), and of the complemented strain *cPRK1* was tested on YPD supplemented with CuCl_2 (10 and 13 mM) or CuSO_4 (13 and 15 mM). The *cup2* deletion mutant was included as a control. Plates are shown after 2 days of incubation at 30 °C. Source data (uncropped photographs) are provided as a Source Data file.

sensitivity to copper, as previously observed in *C. albicans*[37] (Fig. 1B). Disrupting *CUP9* (*CPAR2_301340*), *MAC1* (*CPAR2_301500*) and *PBS2* (*CPAR2_806750*) increases sensitivity to cadmium[41] (Fig. 1B). Disrupting the phosphatidylinositol (PI) 3-kinase *VPS34* (*CPAR2_206880*) also induced hypersensitivity to oxidative and cell wall stresses, copper and cadmium, and azoles (Fig. 1B), as also observed in *C. albicans* and *C. glabrata*[42–44].

Some novel functions were also identified. Not surprisingly, deleting *CUP2*, which in *C. albicans* induces transcription of copper resistance genes[37], results in increased sensitivity to copper (Fig. 1B). However, the same effect is observed when deleting or disrupting *PRK1* (*CPAR2_800700*), which encodes a putative protein kinase. The phenotype is restored when the wild-type gene is reintroduced (Fig. 1C), strongly suggesting that Prk1 is a novel regulator of metal sensitivity.

We also evaluated the ability of the mutant collection to use amino acids as a sole nitrogen source (Supplementary Fig. 3 and Supplementary Data 2, 3). Loss of the Nitrogen Catabolite Repression (NCR)-sensitive Gat1 activator (*CPAR2_500590*) dramatically reduced growth on leucine, methionine, threonine, and tryptophan, as previously shown[45,46]. Disrupting *SSN3* affected the growth of lysine, cysteine, and histidine. Similarly, strains with disruptions or deletions of *bna2*, *prk1*, *rtg3*, and *rfx2* were unable to use lysine as a sole nitrogen source (Supplementary Fig. 3). Deleting the transcription factor *PUT3* (*CPAR2_208790*) prevents cells from using proline as a nitrogen source, as it prevents the expression of *PUT1* and *PUT2*, which convert proline to glutamate[45,47]. The role of Aro80 (*CPAR2_108570*) as a regulator of aromatic amino acid catabolism is conserved between *C. albicans* and *C. parapsilosis*; disruption severely impaired the ability of *C. parapsilosis* to use tryptophan[48], and affected growth on isoleucine, leucine, threonine, and valine. Similarly to what is known for *C. albicans*, disrupting *ARG83* (*CPAR2_400180*) results in the inability to use proline as a nitrogen source[37], and Uga3 (*CPAR2_200790*) is required for using Gamma Amino-Butyric Acid (GABA)[37,45]. Disrupting the arginase-encoding *CAR1* (*CPAR2_100820*) and deleting *PPR1* (*CPAR2_101530*) confirmed that they are required for arginine[45] and allantoin catabolism[47], respectively.

In addition, some potentially novel regulators were identified. Disrupting homologues of the Candida-specific filamentous growth regulator (*FGR*) gene family show that they play a role in nitrogen metabolism in *C. parapsilosis*: deleting *FGR15* (*CPAR2_213110*) resulted in reduced growth on YNB + glutamate, proline or valine, and the Δfgr3 (*CPAR2_302310*) strain could not use urea (Supplementary Fig. 3). Deleting *RBF1* (*CPAR2_600470*) resulted in poor growth on arginine, GABA, glutamate, glutamine, and glycine (Supplementary Fig. 3).

## Regulation of methionine synthesis by Met4 and Met28

Whereas many *C. parapsilosis* genes have the same or similar functions as orthologs in other yeast species, we noticed one major difference when we evaluated the ability of the mutant collection to synthesise amino acids, uracil and adenine (Supplementary Note 2 and Supplementary Fig. 4). Surprisingly we found that *C. parapsilosis* cells lacking orthologs of Met4 could grow in the absence of cysteine and methionine (Figs. 2, 3).

In many yeast species, Met4 orthologs are absolutely required for the synthesis of methionine. For example, disruption of *MET4* in *C. albicans* and *S. cerevisiae* makes cells dependent on externally supplied methionine[32,49], whereas *met4* deletions in the methylotrophic yeast *Ogataea parapolymorpha* require not only exogenous methionine but also cysteine and glutathione[50]. Lack of the Met4 orthologs in the filamentous fungi *A. nidulans* (MetR) and *Neurospora crassa* (Cys3) also results in methionine auxotrophy[33,34].

In *S. cerevisiae*, a paralog of Met4 called Met28 is also required for the regulation of biosynthesis of sulphur amino acids, and for stabilising DNA-bound Met4 complexes[30,31]. Met4 and Met28 proteins are very different, with similarities mostly confined to the basic region and

leucine zipper domain ([28,32], and Fig. 2). Met4 is a much longer protein. It encodes an Activation Domain (AD), a Ubiquitin Interaction Motif (UIM), an Inhibitory Region (IR), a protein-protein interaction domain (INT), an auxiliary domain (AUX), and a degenerated basic region-leucine zipper domain (bZIP) that mediates Met4 dimerisation, and binding to Met28 and Cbf1[29,51] (Fig. 2B). It is not completely clear whether Met28 orthologs are present in species outside *Saccharomyces*. Putative orthologs have been described in *C. albicans* and in *O. parapolymorpha* (Fig. 2[32,50]), but again similarities are restricted to the bZIP regions (Fig. 2).

We identified Met4-like proteins with bZIP regions in yeast species from the CUG-Ser1 clade, the Pichiaceae, and clades within the Saccharomycetaceae[52–54] (Fig. 2A). Met4 orthologs are clearly identifiable; although they range in size, the proteins all encode a bZIP domain, and other domains associated with transcriptional activation and ubiquitin-binding, and protein-protein interaction (INT) (Fig. 2B).

The species also encode a second shorter protein with a bZIP domain. Phylogenetic analysis based on conserved regions only (mostly the bZIP domain) suggests that these are orthologs of *S. cerevisiae* Met28 (Fig. 2A). This supports an ancient origin of the Met4/Met28 paralog pair predating the split between the Saccharomycetaceae/Pichiaceae/CUG-Ser1 clades. The Met4 homologues are more closely related to Met4 from filamentous fungi and *Schizosaccharomyces pombe*. However, the phylogeny is based on a short alignment (69 sites), with relatively poor bootstrap support (< 0.8 on several branches). It is also possible that the "Met28" proteins arose from independent gene duplication events in different species. For example, when the sequences are aligned using Muscle, the inferred tree suggests that Met28 proteins from *Saccharomyces* and related species form a separate clade (Supplementary Fig. 5). For simplicity, however, we will refer to all the shorter proteins with bZIP domains as "Met28".

Like in other yeast species, *C. parapsilosis* Met28 is shorter than Met4, and it lacks most of the known N-terminal and central domains (Fig. 2B). However, their C-terminal bZIP domains are approximately 60% identical, suggesting that they may have overlapping DNA targets. We, therefore, tested the role of *C. parapsilosis* Met28 in the synthesis of sulphur amino acids (Fig. 3). Similar to disrupting *MET4*, disrupting *MET28* alone had no effect on the ability of *C. parapsilosis* to grow in the absence of externally supplied cysteine and methionine (Fig. 3A). However, when both *MET4* and *MET28* are deleted, the strain is auxotrophic. Testing the requirements for each amino acid separately shows that the strain is defective in methionine synthesis (Fig. 3A). The same phenotype was observed whether *MET4* and *MET28* are deleted or disrupted by introducing premature stop codons (Fig. 3). The wildtype phenotype is restored by removing the premature stop codon from either the edited *MET4* (cMET4) or from the edited *MET28* (cMET28), confirming that a wildtype copy of either gene is sufficient to restore the ability to synthesise methionine (Fig. 3A).

Several recent studies have shown that gene disruptions can result in different phenotypes in different isolates of the same species[55–57]. Because our observation that *C. parapsilosis* Met4 is not the sole regulator of sulphur amino acid biosynthesis is surprising, we tested the effect of disrupting both *MET4* and *MET28* in different genetic backgrounds and in isolates from different *C. parapsilosis* clades[58]. The original disruptions are in *C. parapsilosis* CLIB214 (Clade 2). We disrupted both genes in *C. parapsilosis* 02-203 (Clade 5), 81-042 (Clade 3), and CDC179 (Clade 4) (Supplementary Data 1)[58]. In isolates from all clades, disrupting both *MET4* and *MET28* together was required to abolish the ability to grow in the absence of methionine (Fig. 3B).

To determine whether the sulphur regulatory pathway in *C. albicans* or *C. parapsilosis* reflects that of other *Candida* species, we also attempted to CRISPR edit *MET4* and *MET28* in the related species *Candida tropicalis* (CAS08-102, [35]). However, while two independent

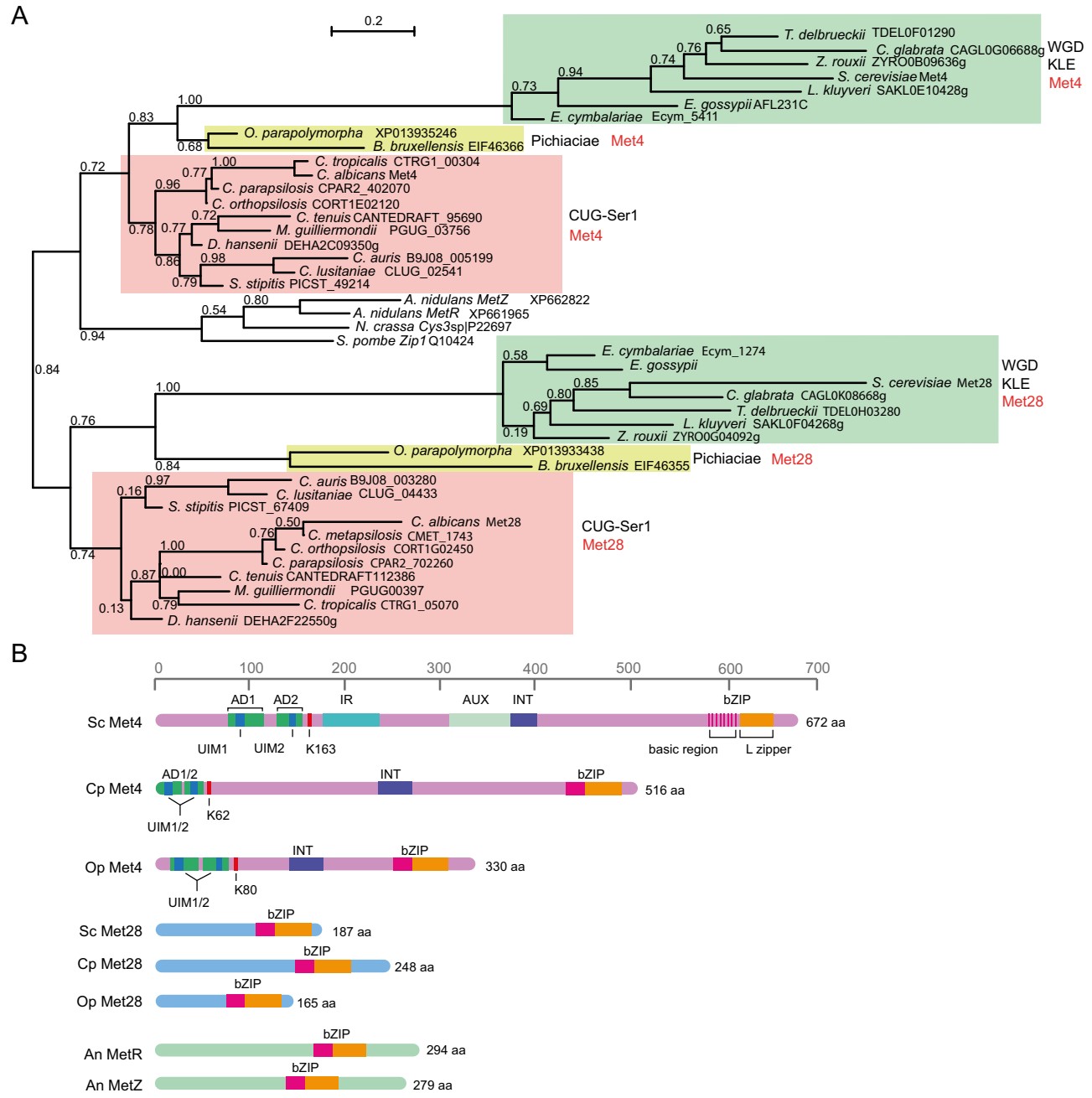

**Fig. 2 | Phylogeny of the Met4 and Met28 paralogs in budding yeasts. A** Protein sequences retrieved from YGOB[52], CGOB[53] or the indicated accession numbers were aligned using ClustalO implemented in Seaview[54]. Trees were inferred using PhyML restricted to conserved regions selected using Gblocks. WGD = Whole Genome Duplication; KLE = *Kluyveromyces/Lachancea/Eremothecium*. **B** Met4 and Met28 sequences of *S. cerevisiae* (Sc), *C. parapsilosis* (Cp) and *O. parapolymorpha* (Op) are drawn to scale. The *A. nidulans* duplicated protein pair MetR/MetZ is also included. Met4-like sequences in yeast species from the CUG-Ser1 clade, the Pichiaceae, and clades within the Saccharomycetaceae share a common domain organisation. Starting from the N-terminal end of the protein: Activation Domains

(AD) encompassing Ubiquitin Interaction Motifs (UIM, which protects ubiquiti-nated Met4 from degradation), a conserved Lysine (K) that is the target of poly-ubiquitination, Inhibitory Region (IR, required for Met30-mediated inhibition of activity in the presence of a high concentration of methionine), Auxiliary domain (AUX, required to fully relieve IR-mediated repression), Interaction domain (INT, required for binding of Met31/Met32), and the bZIP DNA binding domain. The degenerated basic region of the bZIP domain in *S. cerevisiae* Met4 is indicated by the pink vertical stripes. Met28-like sequences are shorter and lack the N-terminal domain organisation observed in Met4, but they have a bZIP binding domain at the C-terminal end.

lineages with an edited *met28* gene were easily obtained, repeated transformations targeting two different protospacer sequences on *MET4* failed to generate the desired edit, suggesting that *MET4* may be an essential gene in *C. tropicalis*. Interestingly, the lack of a functional *MET28* alone was enough to prevent methionine biosynthesis (Fig. 3B), suggesting that the regulatory pathway is different in *C. albicans, C. parapsilosis* and *C. tropicalis*.

## Met4 and Met28 regulate different genes in *C. parapsilosis*

Figure 3 shows that both *MET4* and *MET28* are involved in the regula-tion of methionine biosynthesis in *C. parapsilosis*. To dissect their individual contribution, we used RNA-seq analysis to compare the transcriptional profile of the mutant strains to the parental *C. para-psilosis* CLIB214 in the presence or absence of cysteine and methionine. A visual representation of the data sets is provided in Fig. 4 (and

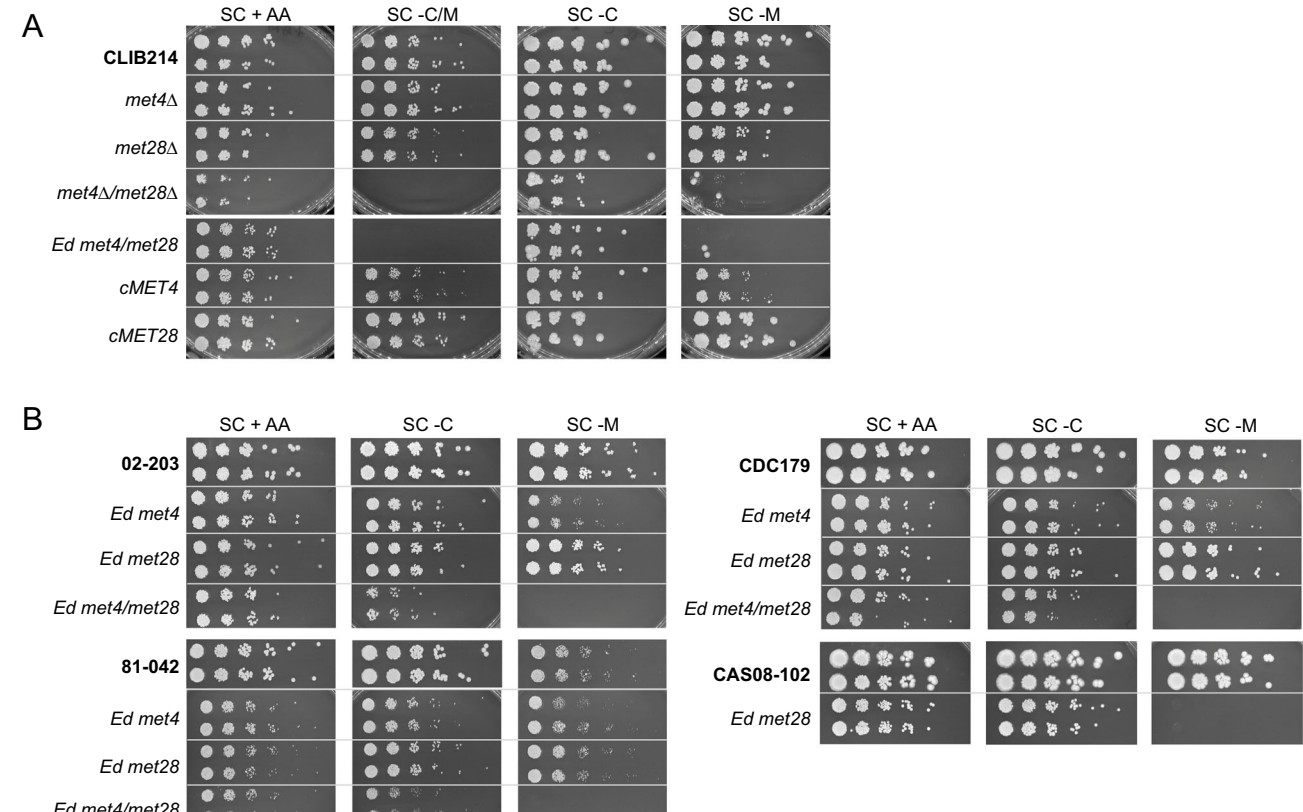

**Fig. 3 | Either Met4 or Met28 can sustain *C. parapsilosis* growth in the absence of methionine. A** *C. parapsilosis* CLIB214 and derivative strains disrupted in *MET4*, *MET28*, or both were tested for their ability to grow in the absence of sulphur-containing amino acids. Cell dilutions were spotted on SC media supplemented with all amino acids (SC + AA), or lacking cysteine/methionine (SC – C/M), cysteine (SC – C) or methionine (SC – M) and photographed after 48 h at 30 °C. Only disrupting both *MET4* and *MET28*, either by deletion or by insertion of a premature stop codon (*met4Δ/met28Δ* and *Ed met4/met28*), abolished methionine synthesis. Restoration of the wildtype sequence of either *MET4* (cMET4) or *MET28* (cMET28) recovered methionine prototrophy. **B** The same result was obtained when the mutations were introduced in three additional *C. parapsilosis* strains from Clades 5 (02−203), 3 (81-042), and 4 (CDC179)[58]. On the contrary, disruption of *MET28* in the related species *C. tropicalis* (CAS08-102) was enough to abolish methionine synthesis. Source data (uncropped photographs) are provided as a Source Data file.

Supplementary Note 3 and Supplementary Fig. 6), and all genes that were differentially expressed under cysteine/methionine starvation in at least one of the strains are listed in Supplementary Data 4.

A total of 109 genes were differentially expressed in wildtype *C. parapsilosis* under Cys/Met starvation, of which 93 are up- and 16 are down-regulated (Fig. 4, Supplementary Data 4 and Supplementary Table 2). Of the 93 overexpressed genes, at least 45 have likely roles in sulphur metabolism (shown in red in Fig. 4 and in more detail in Fig. 5 and Supplementary Table 2). These include sulphur transporters like *SUL2* and oligopeptide/amino acid permeases like *AGP3* and *MUP1*, most enzymes required for the activation and further reduction of sulphate ($SO_4^{2-}$) to sulphite ($SO_3^-$) (*MET3*, *MET14*, and *MET16*), the two subunits of the sulphite reductase (*ECM17*, *MET10*) converting sulphite to sulphide ($S^{2-}$), and the enzymes catalysing the last steps leading to the production of homocysteine (*MET2*, *MET15*, and *MET13*) (Fig. 5). Several genes involved in the methyl cycle (*SAM2*, *MET1*), transsulfuration pathway (*CYS3*, *STR3*), and glutathione synthesis (*GCS1*, *GST2*) are upregulated as well. In addition, expression of the transcription factors *MET4* and *MET32* and of the ubiquitin-responsive regulatory protein *MET30* is induced (Fig. 5, Supplementary Data 4 and Supplementary Table 2). By contrast, the transcriptional level of *MET28* is not affected by cysteine/methionine levels.

Deleting both *MET4* and *MET28* together abolishes sulphur-limitation induction of expression of the 45 likely sulphur-related genes (Fig. 4 and Supplementary Table 2). Of these, fourteen are enzymes implicated in the sulphate assimilation pathway, the methyl cycle and the transsulfuration pathway (Supplementary Table 2: categories Sulphate assimilation, Met metabolism, Cys metabolism; Fig. 5).

Deleting *MET28* alone has a dramatic effect on the expression of these fourteen genes (Figs. 4, 5 and Supplementary Table 2: categories Sulphate assimilation, Met metabolism, Cys metabolism). Expression of eleven of the fourteen is no longer induced by sulphur limitation (i.e., absence of cysteine and methionine) when *MET28* is deleted, suggesting that these genes are targets of Met28. These include genes involved in sulphate assimilation (*MET10*, *MET14*, *MET16*, *MET2*) and methionine/cysteine metabolism (*SAM2*, *MET1*, *MET13*, *STR3*) (Fig. 5 and Supplementary Table 2). Expression of the genes *MET3*, *ECM17* (ortholog of *S. cerevisiae MET5*), and *MET15* (ortholog of *S. cerevisiae MET17*) are still induced in sulphur-limiting conditions, but at a lower extent than in the wildtype CLIB214 strain, suggesting that Met28 plays a sizeable role in regulating their transcription (Fig. 5 and Supplementary Table 2). Most of these genes are typically targets of Met4 in other yeast species, like *S. cerevisiae* (all listed), *C. albicans* (*MET14*, *MET16*, *MET2*, *SAM2*, *STR3*, *MET15*), and *O. parapolymorpha* (*SAM2*, *MET13*, *MET3*, *MET5*)[25,32,50].

In addition, in *C. parapsilosis* Met28 is also required for induction of the enzymes of the O-acetyl-serine (OAS) pathway, an alternative pathway for the synthesis of cysteine from sulphide and O-acetyl-serine that is missing in *S. cerevisiae*, in response to sulphur limitation[28]. The OAS pathway (in green in Fig. 5) requires two enzymes, serine-O-acetyltransferase (which is missing in *S. cerevisiae*) and cysteine synthase[28]. Like other yeasts, *C. parapsilosis* encodes one likely serine-O-acetyltransferase (*CPAR2_203640*) and two likely

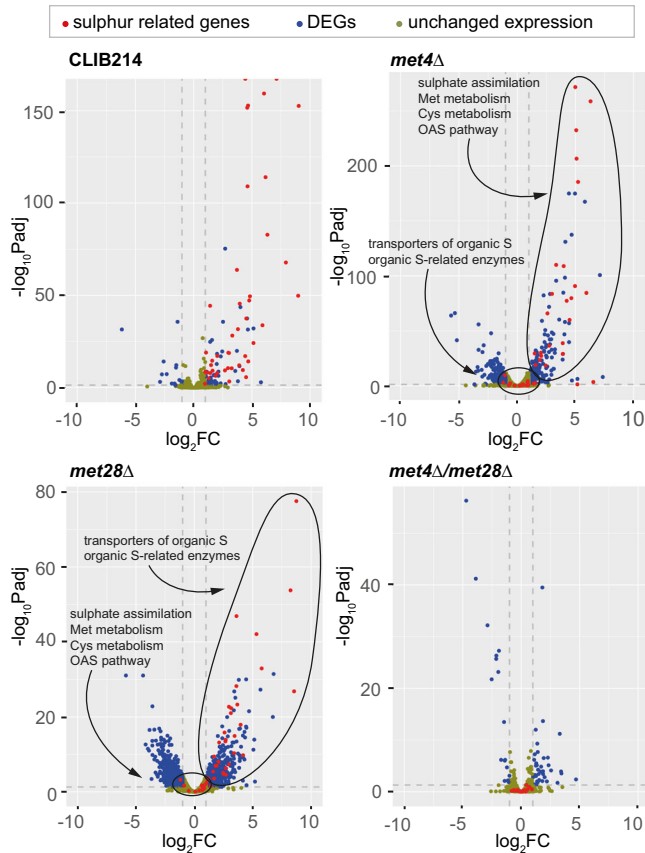

**Fig. 4 | Loss of function of Met4 or Met28 affects a different subset of sulphur-responsive genes in *C. parapsilosis*.** The transcriptional response to cysteine/methionine (C/M) starvation was analysed by RNAseq in *C. parapsilosis* CLIB214 and strains in which *MET4*, *MET28*, or both were deleted. The results are presented in volcano plots (see also Supplementary Data 4, Supplementary Table 2, Supplementary Note 3 and Supplementary Fig. 6). For each gene, the log$_2$ fold change (log$_2$FC) and the -log$_{10}$ of the adjusted *P*-value (-log$_{10}$Padj) are indicated. Values of FC > 2 and adjusted $p < 0.05$ were selected to define differentially expressed genes (DEGs) between the two conditions (absence of C/M versus presence of C/M). Three dotted grey lines delineate the threshold of significance and fold change. Unaffected genes are coloured light brown, while DEGs are blue. Genes that are implicated in sulphur metabolism are highlighted in red (Supplementary Table 2). Deleting *MET4* or *MET28* individually affects the expression of two separate subsets of genes, circled in two clusters in the figure (categories as in Supplementary Table 2 and Supplementary Fig. 6), and the absence of both regulators completely shuts down the induction of sulphur-responsive genes: the expression of all the sulphur-related genes collapses below the threshold of significance/fold change in the *met4Δ/met28Δ* mutant.

cysteine synthases (*CPAR2_300800* and *CPAR2_602080*), of which expression of only one (*CPAR2_300800*) is upregulated in sulphur limiting conditions (Fig. 5).

In contrast, deleting *C. parapsilosis MET4* alone has relatively little effect on the expression of genes required for the biosynthesis of cysteine and methionine; expression of twelve of the fourteen enzymes involved in the process is still induced by sulphur limitation (Supplementary Table 2 and Figs. 4, 5). However, transcription of a number of uncharacterised genes is no longer induced by sulphur limitation (Fig. 5 and Supplementary Table 2). These include several members of the Dal5 allantoate permease family of fungal major facilitator superfamily (MFS) transporters such as *SOA1* (*CPAR2_702960*), which in *S. cerevisiae* mediates the uptake of sulphate (at low affinity) and the uptake of sulphite/thiosulfate/sulfonates (at high affinity)[59], and *CPAR2_200220* and *CPAR2_203280*, which are homologous to *S. cerevisiae SEO1*, a putative sulphur compound transporter[60].

Although the assimilation of alternative sulphur sources by fungi is still relatively unexplored, several classes of gene families were identified in fungal genomes based on their similarity to bacterial genes that are likely involved in this process[27,61]. We observed the induction of expression of many members of these gene families in *C. parapsilosis* upon cysteine/methionine starvation, which was abolished when *MET4* was deleted (Supplementary Data 4, Supplementary Table 2 and Fig. 5). These include enzymes required for the assimilation of organosulfur compounds such as Flavin mononucleotide (FMNH$_2$)-dependent monooxygenases (Pfam accession no. PF00296), 2-oxoglutarate-dependent dioxygenases (PF02668), and arylsulfatases (PF00884).

Flavin mononucleotide (FMNH$_2$)-dependent monooxygenases can oxidate sulfones (R − S(=O)$_2$ − R') (and possibly sulfoxides, R − S(=O) − R') into sulfonates (R − S(=O)$_2$ − O$^-$)[62]. Subsequent cleavage of the remaining S − C bond within the sulfonate group releases the remaining side-chain as an aldehyde and sulphite, which is funnelled into the pathway for cysteine/methionine synthesis. Both FMNH$_2$-dependent monooxygenases and 2-oxoglutarate-dependent dioxygenases can catalyse the desulfonation reaction. Another enzyme class, arylsulfatases (PF00884), can hydrolyse sulphate esters (R − O − SO$_3^-$) to release sulphate and alcohol. This reaction can also be performed in an oxygen-dependent manner by some 2-oxoglutarate-dependent dioxygenases, resulting in the release of sulphate and an aldehyde[62]. Our results show that the expression of four 2-oxoglutarate-dependent dioxygenases, four FMNH$_2$-dependent monooxygenases, and two arylsulfatases is dependent on Met4 (and not Met28) (Supplementary Data 4, Supplementary Table 2 and Fig. 5: dashed box). The fact that the expression of genes with a predicted function in the uptake and processing of organosulfur compounds depends on Met4 suggests that this transcription factor may be required in the assimilation of alternative sulphur sources in *C. parapsilosis*.

Although Met4 and Met28 regulate different subsets of genes, there are also some overlaps. Expression of some genes (e.g., the sulphate permease *SUL2*, the oligopeptide permease *AGP3*, the putative sulphur transporter *SEO1*) requires either Met4 or Met28; expression is abolished when both genes are deleted together, but not when either gene alone is deleted (Supplementary Data 4, Supplementary Table 2 and Fig. 5). In principle, these genes may explain the functional redundancy observed between Met4 and Met28. However, disrupting each individually in *C. parapsilosis* CLIB214 does not result in methionine auxotrophy (Supplementary Fig. 7). It is possible, however, that if all genes were simultaneously deleted, methionine synthesis would be disrupted.

For other genes, expression is more dependent on one transcription factor: examples include the 2-oxoglutarate-dependent dioxygenase *JLP1* and the predicted allantoate permease *CPAR2_702970*, which are mainly regulated by Met4, and the methionine/cysteine permease *MUP1*, which is mainly regulated by Met28 (Supplementary Data 4, Supplementary Table 2 and Fig. 5).

Overall, the transcriptional analysis confirmed that both Met4 and Met28 are major players in regulating the pathway for the assimilation of sulphur and the synthesis of the sulphur-containing amino acids, but it also suggested each transcription factor may have specific roles.

## Promoters of sulphur responsive genes contain Cbf1 and Met32 binding motifs

To try to identify putative binding sites for Met4 and Met28, we used the MEME-suite software to detect conserved motifs in the 1-kb region upstream of the differentially expressed sulphur-related genes (in red in Supplementary Table 2)[63]. Four statistically significant motifs were found: (i) 5′MAAAACTGTGGYKBH3′ (E value 6.9e-068), (ii) 5′TTTTTTYTTTTTTTYTTTTT3′ (E value1.6e-043), (iii) 5′CACACAYA-CACACACACACA3′ (E value 2.2e-019), and (iv) 5′TCACGTGMAWW3′ (E

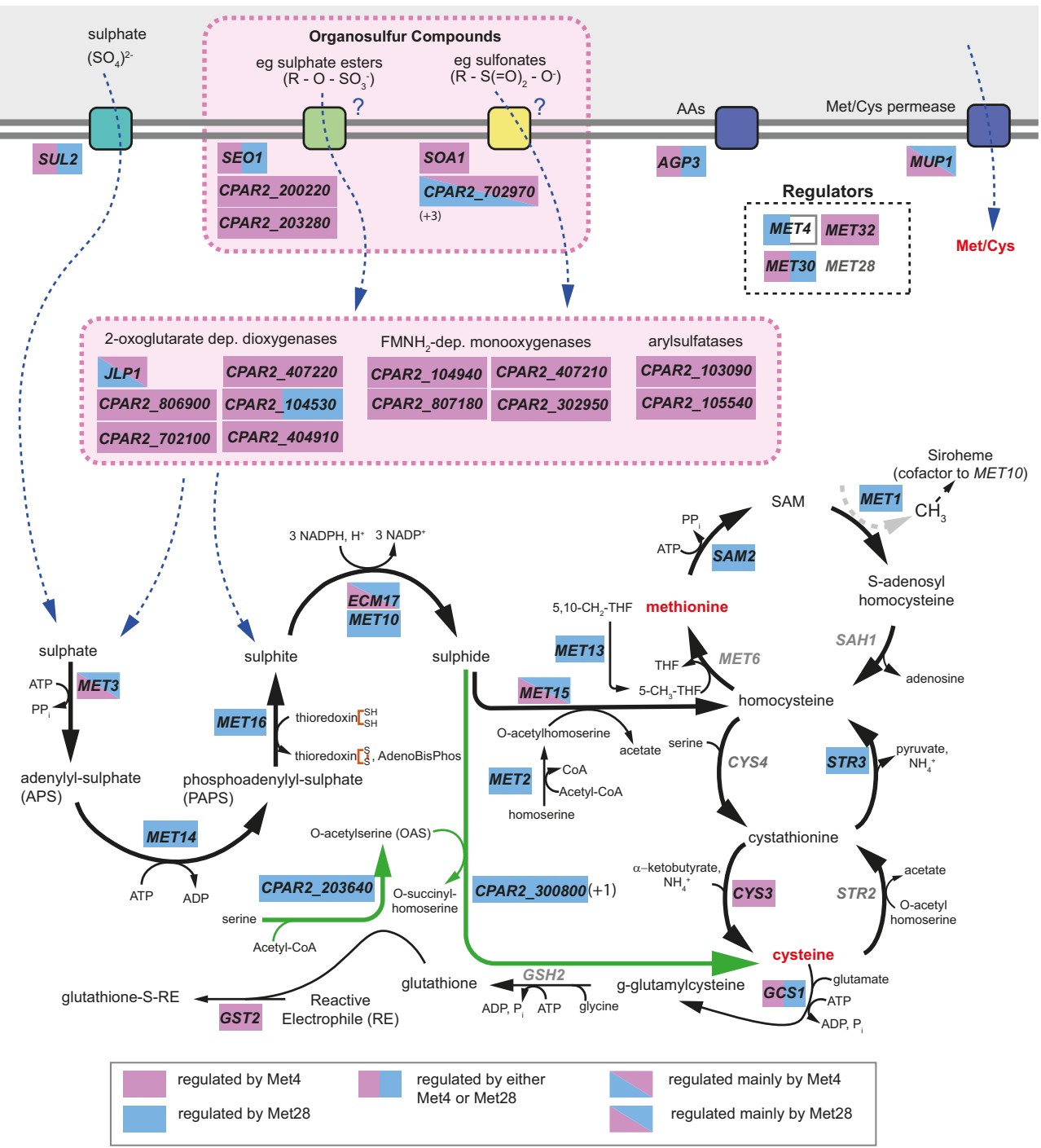

**Fig. 5 | Met4 and Met28 regulate different branches of sulphur metabolism in *C. parapsilosis*.** Sulphur-responsive genes based on the transcriptomic analysis, and their involvement in the import and assimilation of sulphur sources. All the genes written in black are upregulated in response to cysteine/methionine starvation in *C. parapsilosis* CLIB214. The expression of the genes in grey is not significantly affected by the absence of cysteine and methionine. The DEGs are boxed in coloured rectangles to incorporate information on how deleting *MET4*, *MET28*, or both affected the network that is controlled by these regulators (Supplementary Data 4 and Supplementary Table 2). The genes are boxed in blue if their expression is induced by Met28, and in pink if they are controlled by Met4 (Supplementary Data 4 and Supplementary Table 2). A split rectangle indicates that both regulators participate (see legend below the pathway). We could not determine if *MET4* regulates itself because our data was obtained in a *met4Δ* strain: for this reason, half of the rectangle was left white. The pathway of inorganic sulphur assimilation into

methionine (methyl cycle) and cysteine (transsulfuration pathway) through sequential reduction of sulphate into sulphite and then sulphide is conserved between *C. parapsilosis* and *S. cerevisiae*, but in *C. parapsilosis* the genes involved are mainly controlled by Met28. Met28 also controls the O-Acetyl-Serine (OAS) pathway (in green), which leads to the synthesis of cysteine from serine and sulphide and is absent in *S. cerevisiae*. On the contrary, Met4 is responsible for the expression of genes with a predicted role in the import and assimilation of organosulfur compounds (e.g., sulphate esters or sulfonates) (highlighted by the dotted-lined pink rectangle). The numbers in parenthesis indicate the presence of additional members of a gene family in the genome of *C. parapsilosis*, which are not overexpressed in the absence of Cys/Met in our dataset. The boxes representing the transporters are colour-coded based on their putative specificity for sulphate (teal), organosulfur compounds (green and yellow), or amino acids (purple). AdenoBisphos = adenosin 3′,5′-bisphosphate; THF = tetrahydrofolate.

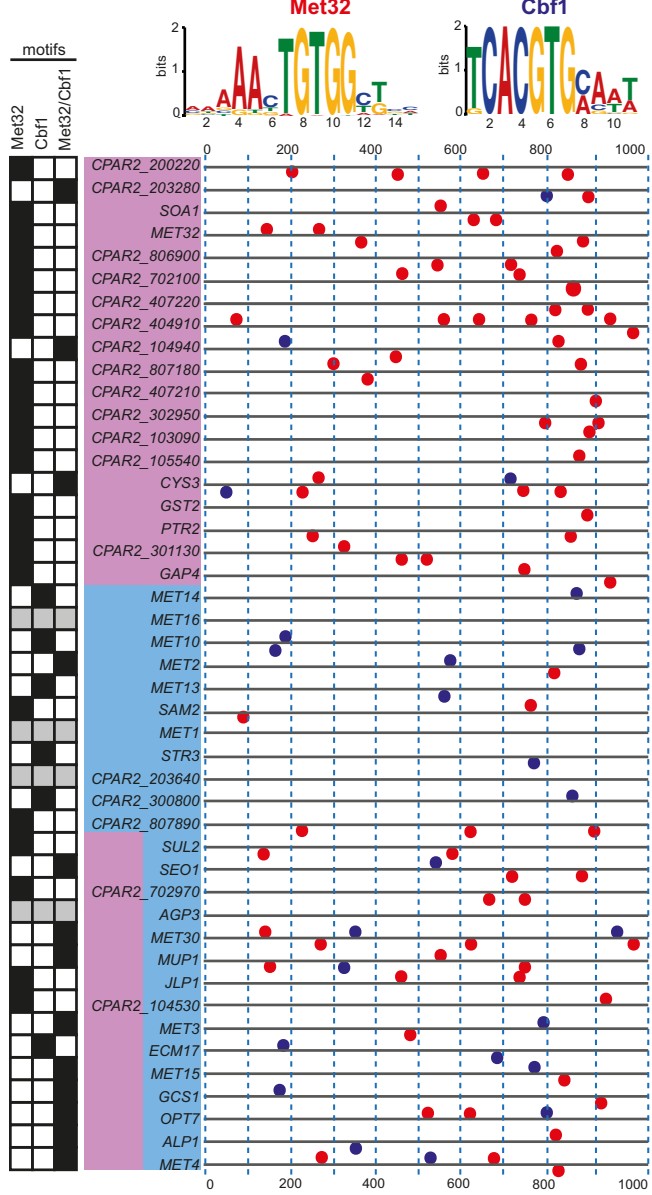

**Fig. 6 | *C. parapsilosis* sulphur responsive genes contain Cbf1 and Met32 binding sites.** The promoter regions (1-kb regions upstream of the starting codon) of the genes upregulated in the absence of Met/Cys were analysed using MEME-suite[63]. The promoters are depicted as lines (0 = −1000bp from ATG; 1000 = −1bp from ATG). The genes are listed on the left, colour-coded based on their dependency on Met4 (pink) or Met28 (light blue). Two motifs were found, corresponding to sites that are known to mediate the DNA binding of Met32 and Cbf1 in *S. cerevisiae* (illustrated above the graph). The distribution of the motifs is indicated by red (Met32) or blue (Cbf1) dots, located above (+ strand) or below (− strand) the promoter lines. The precise coordinates are provided in Supplementary Table 3. The promoters contain Met32 only, Cbf1 only, or both binding sites (indicated by the motif checkboard on the left-hand side; grey box: no motifs found with the selected stringency on MEME).

value 1.2e-002) (Fig. 6). The 5'TCACGTGMAWW3' motif is very similar to the binding site for the basic helix-loop-helix (bHLH) transcription factor Cbf1, both in *C. albicans*[32] and *S. cerevisiae*[27]. The 5'MAAAA<u>CTGTGG</u>YKBH3' motif closely resembles the Met31/Met32 binding site (5' <u>CTGTGGC</u> 3') found in *MET* promoters in *S. cerevisiae*[27], and it partially overlaps with the proposed Met4 motif in *C. albicans* (5'GTWGT<u>RGTGG</u>3')[32]. We refer to these sequences as Cbf1 and Met32

binding motifs, respectively, from now on. It is very unlikely that the highly repetitive motifs (ii and iii) have any biological significance.

MAST analysis shows that at least one of the Cbf1 and Met32 binding motifs is present in 41/45 promoters analysed: 35/41 promoters contained single or multiple copies of the Met32 motif, whereas 19/41 promoters contained single or multiple copies of the Cbf1 motif (Fig. 6 and Supplementary Table 3). In 22/41, only Met32 motifs and in 6/41, only Cbf1 motifs were identified, and both motifs are present in 13/41 promoters. The Met32 binding motifs are enriched in the promoters of genes regulated either exclusively or partially by Met4, based on our transcriptional analysis (Fig. 6). In contrast, Cbf1 motifs are present in genes regulated by Met28 alone, and by both Met4 and Met28 (Fig. 6). We did not identify any Met32 or Cbf1 significant binding motifs upstream of *MET16*, *MET1*, *CPAR_203640*, and *AGP3*.

Our findings suggest that the transcriptional effects of Met4 and Met28 are likely achieved through association with Met32 and Cbf1, respectively. This hypothesis is supported by the observation that disrupting *MET28* affects both the expression of sulphur-related genes, and also the expression of genes involved in the insertion of proteins into the mitochondrial inner membrane, assembly of the mitochondrial cytochrome c oxidase, glycolytic and glucose metabolic processes, cytoplasmic translation, and ribosomal small subunit biogenesis (among others) (Supplementary Data 4, GO enrichment analysis tab)[64]. Genes involved in these biological processes are also the target of Cbf1 in *C. albicans*, in which this transcription factor is not only involved in sulphur metabolism, but also in respiration, glycolysis, and expression of ribosomal proteins[65]. We were unable to disrupt *CBF1* in *C. parapsilosis* suggesting that it is essential, so we could not directly test the effect on phenotype.

In *S. cerevisiae*, the zinc finger proteins Met31 and Met32 recruit Met4 to the promoter of sulphur-responsive genes[25]. In *C. parapsilosis* there is only one *MET31/32* homologue, the zinc finger transcription factor *MET32* (*CPAR_407160*). The large-scale screen showed that deleting *MET32* does not cause cysteine/methionine auxotrophy (Supplementary Data 3), which was further confirmed by serial dilution assays (Fig. 7B). However, the transcriptional profiling showed that expression of *MET32* is induced by sulphur limitation, and it is dependent on Met4 and not Met28 (Supplementary Data 4, Supplementary Table 2 and Fig. 5). The role of *MET32* was, therefore, further explored.

## Met4 and Met32 regulate the assimilation of organic sulphur

Expression analysis (Fig. 5) suggests that Met4 may be required for the assimilation of alternative sulphur sources in *C. parapsilosis*. We, therefore, tested the effect of deleting *met4* and *met28* on growth using either ammonium sulphate (AS) or alternative organosulfur compounds as the sole sulphur sources. Three different kinds of organosulfur compounds were tested: (i) (3-(N-morpholino) propanesulfonic acid) (MOPS) (sulfonate); (ii) dimethyl sulfoxide (DMSO) (sulfoxide), and (iii) sodium dodecyl sulphate (SDS) (sulphate ester) (Fig. 7). As a previous study showed that *Lodderomyces elongisporus* - a close relative of *C. parapsilosis* in the CUG-Ser1 clade - displayed robust growth using organosulfur compounds, we included *L. elongisporus* CBS2605 as positive control (Fig. 7A)[61].

*C. parapsilosis* CLIB214 can use both inorganic and organic sulphur to sustain growth, which was not affected by deleting *MET28* (Fig. 7A). However when *MET4* is deleted, cells are unable to grow when organosulfur compounds are the sole sulphur source, but can grow when inorganic sulphur (ammonium sulphate, AS) is provided (Fig. 7A). The wildtype phenotype is restored when the premature stop codon disrupting *MET4* is removed (cMET4) (Fig. 7). *C. parapsilosis*, therefore, requires Met4 to use MOPS, SDS, or DMSO as sole sulphur sources.

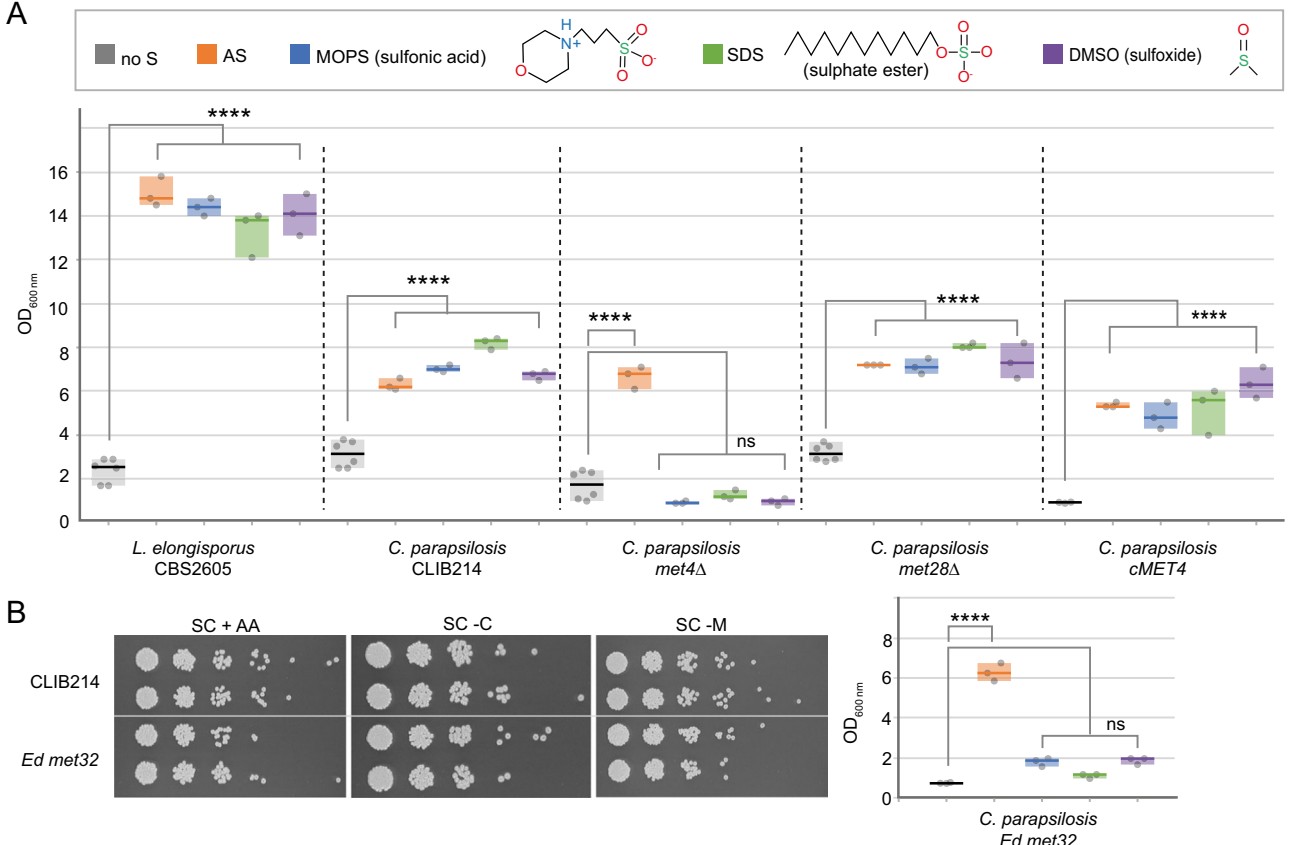

**Fig. 7 | Met4 and Met32 are required for the assimilation of organosulfur compounds in *C. parapsilosis*.** The growth of *C. parapsilosis* CLIB214 and derivative strains disrupted in *MET4*, *MET28*, or *MET32* was tested in the presence of sulphur-free media supplemented with (i) no sulphur (control, no S); (ii) inorganic sulphur (sulphate, AS), or (iii) organosulfur compounds from different structural classes as only available sulphur source (MOPS, SDS, DMSO). The $OD_{600}$ of the cultures after 8 days at 30 °C is shown. Individual observations are plotted, and the median is indicated by the horizontal bars. The box represents the range of data. For each condition, 3 biological replicates were tested ($n = 3$), except for *L. elongisporus* CBS2605 and *C. parapsilosis* CLIB214, *met4Δ* and *met28Δ* in the absence of sulphur, in which 6 biological replicates were tested ($n = 6$). Detailed information on the mean and standard deviation is available in the Source Data file. One Way ANOVA and Tukey HSD test ($F = 222$, degrees of freedom = 29) were used for statistical analysis. ns = not significant, ****$P \leq 0.0001$. The exact *P*-values are provided in the Source Data file. *L. elongisporus* CBS2605 was included as positive control[61]. **A** *C. parapsilosis* CLIB214 can use both inorganic and organic sulphur sources. Whereas either Met4 or Met28 can sustain growth on inorganic sulphur (AS), Met4 is required for assimilating the organosulfur compounds tested, as shown by the fact that deleting this gene prevented growth on MOPS, SDS, and DMSO. Restoring the wildtype sequence in a *met4*-disrupted strain recovered this ability (*cMET4*). **B** Growth in the absence of cysteine or methionine confirmed that the transcription factor Met32 is not required for the synthesis of sulphur-containing amino acids. However, this protein – like Met4 – is also required for assimilating organic sulphur, as disruption of *MET32* specifically abolished growth in media containing organosulfur compounds. Source data (uncropped photographs) are provided as a Source Data file.

*C. parapsilosis* cells in which *MET32* was disrupted also fail to grow when MOPS, DMSO, or SDS are the sole sulphur sources, similar to the phenotype of the *met4* deletion (Fig. 7B). Growth on ammonium sulphate is not affected (Fig. 7B). Overall, our results show that Met4 and Met32 are required for assimilation of organic sulphur sources but not inorganic sulphate in *C. parapsilosis*.

## Met28, but not Met4, is required for biofilm formation

*C. parapsilosis* can grow rapidly in total parenteral nutrition and form tenacious biofilms on medically implanted devices[3]. A plethora of different stimuli triggers biofilm formation in *C. albicans,* including high (physiological) temperature, neutral pH, CO₂, and amino acids (including Met)[66]. Hence, we evaluated the contribution of the Met4- and Met28-driven branches of sulphur metabolism to biofilm formation.

We tested the ability of *C. parapsilosis* CLIB214 and strains in which *MET4*, *MET28*, or both were disrupted to form biofilm at 37 °C in media containing high glucose in the presence and absence of methionine[19]. In the presence of methionine, loss of both *MET4* and *MET28*, but not either individually, significantly reduces biofilm formation (Fig. 8). In the absence of methionine, the wild-type strain and cells lacking *MET4* produce comparable biofilms, though overall biofilm is reduced compared to when methionine is present. However, the loss of *MET28* alone completely abolishes biofilm formation, reducing it to the level of the double disrupted strain. Notably, the *MET4*/*MET28* double disruption strain is auxotrophic for methionine, and the *MET28* disruption is not.

## Discussion

Fungal cells can assimilate inorganic sulphur (sulphate) into a variety of sulphur-containing molecules that are essential for life, and they do so using metabolic pathways that are not conserved in mammalian cells[24]. Here, we illustrated the utility of a collection of gene knockouts to dissect the regulation of cysteine/methionine synthesis in the opportunistic pathogen *C. parapsilosis*. Surprisingly, although the bZIP transcription factor Met4 is usually the key regulator of this process in the ascomycetes, we show that Met28 is the core regulator in *C. parapsilosis*. We also find that Met28 is a core regulator of biofilm formation during sulphur limitation, suggesting that it is an important virulence factor.

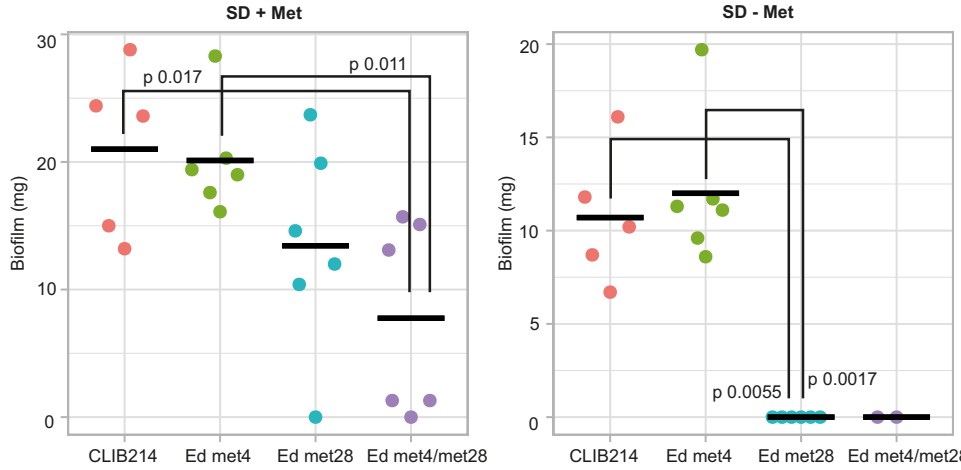

**Fig. 8 | Loss of function of Met28 abolishes biofilm formation in *C. parapsilosis* in the absence of methionine.** Biofilms were formed in SD media with or without Met at 37 °C for 72 h, and then the biomass was weighed. Each dot represents an independent measurement. The bar represents the mean. The exact number of biological and technical replicates and the standard deviation can be found in the Source Data file. Statistical significance was calculated using the Kruskal Wallis test followed by the post-hoc Dunn's test. Significant comparisons ($p < 0.05$) are shown. Source data are provided as a Source Data file.

Much of what is known about the pathways for importing and assimilating sulphate into cysteine and methionine in yeast is based on analysis in *S. cerevisiae*, where it is controlled by a network of transcriptional activators (Met31/32, Cbf1, and Met28) centred around the main regulator Met4[25] (Fig. 9). ScMet4 is unusual in that an insertion in the bZIP region has disrupted its ability to bind DNA[28]. DNA binding is achieved either through interaction with Met31/32 or with a separate DNA binding protein Cbf1[25,32] (Fig. 9A). In *S. cerevisiae* Met4 also interacts with its paralog Met28[30,31]. It is not clear whether Met28 orthologs are found outside the immediate neighbours of *S. cerevisiae* (Fig. 2 and Supplementary Fig. 5). However, small proteins with similar bZIP domains (which we refer to as Met28) are clearly identifiable. Unlike *C. albicans*[32] and *O. parapolymorpha*[50], in *C. parapsilosis* either Met28 or Met4 is required for methionine biosynthesis (Fig. 3), suggesting that the two proteins share some degree of overlapping function. Nonetheless, transcriptional profiling in response to cysteine/methionine starvation clearly showed that Met28 is the main regulator of genes in the pathway (Supplementary Data 4, Supplementary Table 2 and Fig. 5).

The transcription factor Cbf1 regulates methionine biosynthesis in both *S. cerevisiae* and *C. albicans*[25,32]. The presence of conserved Cbf1 binding motifs in the promoters of the genes that are induced during sulphur starvation suggests that this protein plays a role in sulphur metabolism in *C. parapsilosis* (Fig. 6). Cbf1 motifs coincide with genes that are regulated by Met28, suggesting that the proteins interact.

Because the bZIP domain of Met4 is intact in *C. parapsilosis*, we anticipated that it would directly bind to promoter sequences. However, we could not identify likely binding sites in the promoters of regulated genes. Instead, we identified Met32 motifs in genes predominately regulated by Met4 (Fig. 6). This is consistent with Met4 either interacting with or being recruited by Met32 to specific promoters. To our knowledge, there is no direct evidence that Met4 from *S. cerevisiae* or *C. albicans* can bind DNA directly, whether or not the binding domain contains an insertion sequence.

Similar to *S. cerevisiae*, we identified roles for Met4, Met28, Met32 and Cbf1 in the *C. parapsilosis* sulphur regulon. However, there are substantial differences between *S. cerevisiae* (Fig. 9A) and *C. parapsilosis* (Fig. 9B). In *S. cerevisiae*, the genes involved in sulphur metabolism can be divided into three classes[27]: those that show a strict (class 1) or intermediate (class 2) dependency on Cbf1 and Met28, and genes whose transcription depends only on Met31/32 (class 3) (Fig. 9A, Cbf1-Met28 dependent genes and Met32-only dependent genes)[27]. Depending on the promoter sequence, Met4 can be recruited by

Met31/32 to high-affinity Met31/Met32 binding sites (class 2 and 3 genes), or by Cbf1 and Met28 to variant recruitment sites (class 1 genes)[67]. Deleting *CBF1* or *MET28* prevents the expression of class 1 and 2 genes only, whereas the absence of Met4 completely abolishes the induction of the entire regulon. Deleting *MET31/32* also disrupts the expression of all three classes, including class 1, because *MET28* is itself a class 2 gene[27]. The genes required for uptake and assimilation of inorganic sulphur are mainly dependent on Cbf1-Met28 (class 1 and 2), whereas the genes required for the synthesis of homocysteine, the methyl cycle, and assimilation of organosulfur compounds rely on Met31/Met32 (class 3). Genes in the transsulfuration pathway are found in both class 2 and class 3 (Fig. 9A)[27].

In *C. parapsilosis* based on transcriptional analysis and identification of promoter motifs, we propose that there are also three classes of genes in the sulphur regulon: class 1 genes regulated by Met28 and Cbf1, class 2 genes regulated by Met4, Met28, Cbf1 and Met32, and class 3 genes regulated only by Met4 and Met32 (Figs. 5, 9B). Unlike in *S. cerevisiae* and in other fungi, loss of Met4 has no impact on the expression of class 1 genes and little impact on the expression of class 2 genes, as long as Met28 is present, and the cells can still synthesise methionine from inorganic sulphur (Figs. 3, 5, 7, 9B). This suggests that in *C. parapsilosis* Met28 can directly regulate gene expression without requiring an additional activator. As a result, in the absence of Met4 *C. parapsilosis* cells can still import sulphate and synthesise homocysteine and methionine. They can also synthesise cysteine through the OAS pathway, which is absent in *S. cerevisiae* (Fig. 9B). However, disrupting Met4 abolishes the expression of class 3 genes, preventing the assimilation of organosulfur compounds (Figs. 9B, 7).

Notably, the transcription of some of the regulators also differs between *S. cerevisiae* and *C. parapsilosis*. In *S. cerevisiae*, transcriptional levels of *MET31/32* are unaffected by changes in methionine abundance, whereas *MET28* (class 2) expression is induced by the Cbf1-Met28-Met4 complex during sulphur starvation[27]. The converse is true in *C. parapsilosis*: *MET28* is unaffected by methionine levels, and expression of *MET32* is regulated by Met4-Met32 (class 3) (Fig. 5, Supplementary Data 4 and Supplementary Table 2). In both species, *CBF1* expression is unaffected by methionine levels[27]. As a result, the components of the regulatory network are more interdependent in *S. cerevisiae*, where a lack of Met4 and Met32 shuts down the expression of *MET28*. In *C. parapsilosis*, the fact that *MET28* and *CBF1* expression is not dependent on Met4 means that class 1 genes can be expressed in the absence of Met4. The central role of Met28 prompts the question:

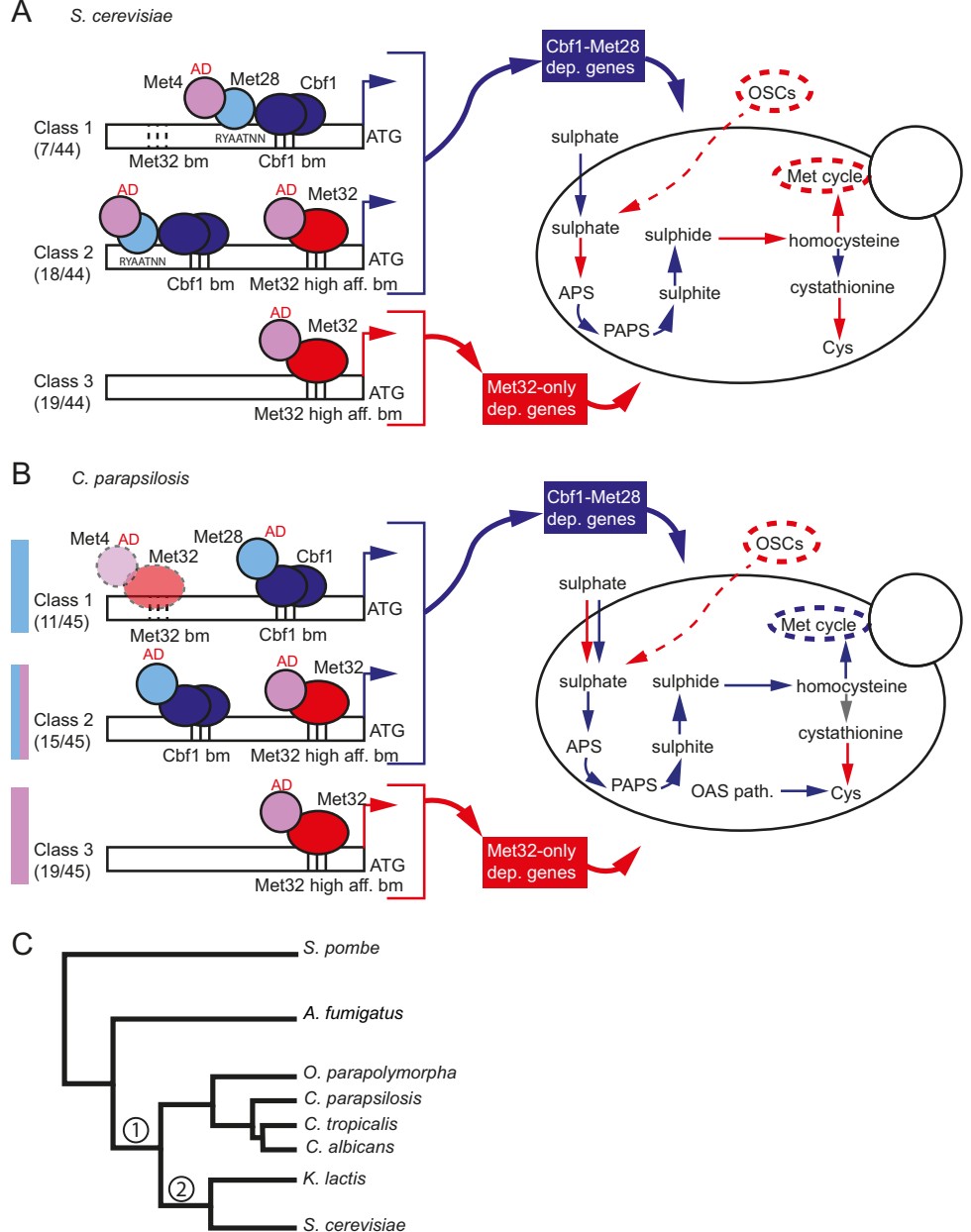

**Fig. 9 | Sulphur metabolism in *S. cerevisiae* and *C. parapsilosis*. A** The expression of the *MET* regulon (44 genes) in *S. cerevisiae* depends strictly (class 1) or more loosely (class 2) on the interaction of Met4 with Cbf1-Met28 (Cbf1-Met28 dependent genes) or Met32 only (class 3, Met32-only dependent genes)[27,67]. The fractions indicate the number of genes in each class, based on ref. 27. Met4 can be recruited by Met31/32 to high-affinity Met31/Met32 binding sites (class 2 and 3 genes), or by Cbf1-Met28 to variant recruitment sites, in which a Met4 recruitment motif (RYAAT) is found 2 bp upstream of the Cbf1 E-box sequence (CACGTG), and potentially recognised by Met28 (class 1 genes)[67]. The yeast vignette shows the different branches of sulphur metabolism that are controlled by genes belonging to Cbf1-Met28 dependent (in blue) and Met32-only dependent (in red) genes, or by genes that are not differentially expressed in sulphur starvation (in grey). AD = activation domain. OSCs = organosulfur compounds. Dotted lines at Met32 binding motifs (bm) in class 1 promoters indicate low-affinity sites. **B** Based on our transcriptional and promoter analyses, the same three classes can be identified in the *MET* regulon (45 genes) in *C. parapsilosis*. However, while Met4 drives the expression of class 3 genes when in association with Met32, Met28 is the core regulator of class 1 genes, possibly recruited by Cbf1. Class 2 genes are targets of both transcriptional complexes and both Met4 and Met28 (Fig. 5). In the absence of *MET28*, Met4 may allow basal expression of class 1 genes through weak binding to low-affinity Met32 binding motifs in promoters of class 1 genes (dotted-lined Met4-Met32 complex). The distribution of Cbf1-Met28 dependent and Met32-only dependent genes in the different branches of sulphur metabolism is not completely conserved between *C. parapsilosis* and *S. cerevisiae* (yeast vignette). **C** Schematic phylogenetic tree showing the relation of the species mentioned in the study. Met28 and Met31/32 were recruited to the sulphur regulatory network in the common ancestor of all budding yeasts (node 1) with Cbf1, and not at a later stage only in *Saccharomyces* and related species (node 2), as previously thought[32].

how can cells lacking *MET28* complete the synthesis of methionine in *C. parapsilosis*? One possibility is that in *C. parapsilosis* (and not in *S. cerevisiae*), weaker binding of the Met4/Met32 complex to low affinity Met32 binding sites in promoters of class 1 genes could allow expression in the absence of Met28.

The circuit regulating methionine biosynthesis has become increasingly complex from filamentous fungi to yeasts; one single regulator (sometimes duplicated, like in *A. nidulans*[33],) is required in filamentous fungi, at least two regulators are required in *C. albicans* (*MET4, CBF1*[32],), and five regulators are involved in *S. cerevisiae* (*MET4,*

*CBF1, MET28, MET31/32*[25]). This led to the hypothesis that regulation evolved in a two-step process; *CBF1* was first recruited in the common ancestor of all budding yeasts (node 1 in Fig. 9C), and then *MET28* and *MET32* transitioned from functioning as general regulators to being involved in methionine synthesis in *Saccharomyces* and related species (node 2, 9 C)[32]. However, we have shown that *MET4, MET28* and *MET32* are involved in methionine synthesis and utilisation of organosulfur compounds in *C. parapsilosis*. In *C. tropicalis, MET28* is also required for methionine synthesis (Fig. 3). The recruitment of *MET28* and *MET32* to sulphur assimilation, therefore, predates the split between the CUG-Ser1 clade and other yeasts (Fig. 9C, node 1). In *C. albicans*, cells lacking *MET32* are still methionine prototrophs, but nonetheless, this gene may be involved in the regulation of organosulfur compound assimilation in this species also[32].

Although most research in fungi has focused on the assimilation of inorganic sulphur (sulphate), sulphur is predominantly available in aerobic soils – the habitat of many fungi – as organosulfur compounds like sulphate esters and sulfonates[68]. The pathways used to scavenge organic sulphur in fungi are currently unknown. However, the process is well studied in bacteria, and comparative studies have identified promising fungal orthologs[62]. We found that *C. parapsilosis* can use organosulfur compounds as sole sulphur sources as efficiently as it uses ammonium sulphate (a source of inorganic sulphur) (Fig. 7B). Assimilation of organic sulphur relies on the expression of putative transporters and desulfonation/oxidation/hydrolysis enzymes, such as flavin mononucleotide dependent monooxygenases, 2-oxoglutarate-dependent dioxygenases and arylsulfatases. Expression of these genes is strikingly increased during sulphur starvation, with $\log_2$ fold changes as high as 9 (Supplementary Data 4 and Supplementary Table 2). We find that Met4 is the major regulator of this sizeable array of transporters, and deleting *MET32* perfectly recapitulates the phenotype of the *met4* deletion (Supplementary Table 2, Figs. 5, 7). This is consistent with Met4 and Met32 interacting to bind to promoters (through the Met32 bZIP domain) and activating transcription (through the Met4 activation domain).

The array of organosulfur compounds that *S. cerevisiae* (and other Saccharomycetaceae) can efficiently utilise is much narrower than yeasts in the CUG-Ser1 clade[61]. This is also reflected in the lower number of enzymes with a predicted role in alternative sulphur sources assimilation found in *S. cerevisiae* compared to, for example, *L. elongisporus* (5 and 13, respectively)[62]. In *S. cerevisiae,* only three genes have been implicated in the assimilation of alternative sulphur sources: the transporter *SOA1*[59], the sulfatase *BDS1*[69], and the sulfonate dioxygenase *JLP1*[70]. These genes are controlled by Met4 in *S. cerevisiae*[27]. In *O. parapolymorpha*, a member of the Pichiaceae, Met4 also controls the expression of some organic sulphur assimilation enzymes, though fewer than in *C. parapsilosis*. Further studies are warranted to determine the importance of organosulfur compounds for the survival of *C. parapsilosis* both in the environment and in the human intestinal flora because organosulfur compounds are the prevalent sulphur source available in both these niches[68,71].

## Methods

### Strains and growth conditions
*C. parapsilosis* and *C. tropicalis* strains (Supplementary Data 1) were grown in YPD medium (1% yeast extract, 2% peptone, 2% glucose) at 30 °C. For colony selection, 2% agar was added. To select for transformants, nourseothricin (Werner Bioagents Jena, Germany) was added to YPD agar at a final concentration of 200 µg/ml. The strains generated in this study were added to the pre-existing collection of *C. parapsilosis* mutant strains[19] and stored in YPD + 30% glycerol in two sets (one per independent lineage) of four 96-well plates.

The library of *C. parapsilosis* was tested for the ability of the strains to (i) grow in the presence of antifungal drugs and compounds affecting the cell wall or inducing osmotic/oxidative stress; (ii) utilise

different nitrogen sources and (iii) synthesise different amino acids (AAs) or adenine.

The phenotypic screens were performed using three different types of media (Supplementary Data 2, 3): (i) YPD agar [1% yeast extract, 2% peptone, 2% glucose, 2% agar] supplemented with different chemical stressors; (ii) Synthetic Defined (SD) media [0.19% YNB without ammonium sulphate (AS) and amino acids (AA), 2% glucose, 2% agar] supplemented with 0.5% AS or 10 mM alternative nitrogen sources; and (iii) Synthetic Complete (SC) media [0.19% YNB without AS and AAs, 2% glucose, 0.5% AS, 2% agar] supplemented with 0.2% complete AAs mix or drop-out mix. All growth data is available in Supplementary Data 2, 3.

### Serial dilution assay
Spot assay experiments were performed to evaluate the growth of selected mutants on Synthetic Complete (SC) media supplemented with 0.2% complete AAs mix or drop-out mix lacking cysteine/methionine, cysteine, or methionine.

Overnight cultures grown in YPD at 30 °C with shaking were washed and diluted to final $OD_{600}$ 0.0625 in 1 ml of PBS. The cultures were serially diluted (1:5) in a 96-well microtiter plate and then spotted on phenotyping plates with a 48-pin bolt replicator. The plates were grown at 30 °C for two days and then photographed.

### Sulphur utilisation growth assays
Growth on different sulphur compounds (inorganic and organic) was tested using a sulphur-limited glucose medium containing only trace amounts of sulphate (6 µM or less) (SLD, 1.2 g yeast nitrogen base/L without amino acids, ammonium sulphate or magnesium sulphate (Formedium, CYN2802), 4 g ammonium chloride/L, 0.84 g magnesium chloride hexahydrate/L and 20 g glucose/L) and supplemented with the chosen sulphur source at 0.1 mM final concentration as in ref. 61.

The strains were pre-cultured overnight at 30 °C in 3 mL minimal glucose medium (MMD, 6.7 g Difco yeast nitrogen base/L without amino acids and 20 g glucose/L), then washed twice with sterile deionized water. The cells were resuspended in 2.97 mL SLD to a final $OD_{600}$ of 0.005 in a 50 mL tube, and individual sulphur compounds were added as 30 µL of a 10 mM stock solution (final concentration 0.1 mM). Chloramphenicol was also added (15 µg/mL) to prevent bacterial contamination. Samples were incubated with shaking at 30 °C for 8 days, and then the $OD_{600}$ was measured. Although 6 days of incubation was sufficient for robust growth of *L. elongisporus* CBS2605[61], the $OD_{600}$ was measured after 8 days of incubation to facilitate the slower growth of *C. parapsilosis* CLIB214. Each strain was tested at least in triplicate.

### Determination of biofilm biomass
The amount of biofilm produced by different strains in the presence or absence of methionine was measured as dry weight, as described in ref. 19, with minor modifications. Cells were grown overnight in YPD at 30 °C with gentle agitation, washed twice in phosphate-buffered saline (PBS) and resuspended at $OD_{600}$ ~ 1 in Synthetic Defined (SD) media [0.19% YNB without AS and AAs, 50 mM glucose, 0.5% AS] supplemented with 10 mg/L His, 20 mg/LTrp and 20 mg/L Met, or SD media supplemented with 10 mg/L His and 20 mg/LTrp only. The cells were seeded in Nunc™ treated petri dishes (10508921, 11 mL/plate) and incubated on a rocker with gentle mixing at 37 °C for 3 h. The plates were then washed once with 1 ml of PBS to remove non-adherent cells, after which fresh media was added. The cultures were incubated in the same conditions for a further ~ 70 h. After washing with PBS, the biofilms were scrapped from the bottom of the plates; the contents of two duplicate plates were vacuum filtered over a 0.8 mm nitrocellulose filter (Millipore). The filters were dried at 37 °C and weighed after 24 h. The average total biomass of each strain was calculated for at least 2 independent samples by subtracting the weight of the filter from the final weight.

## Gene deletions and disruptions

253 *Candida parapsilosis* genes were targeted, including 125 transcription factors (TFs), 70 predicted protein kinases (PKs), 20 genes of unknown function, and 35 genes from chromosome 1. Two different systems were used to generate mutant strains: a fusion PCR method[19] and a CRISPR-Cas9-based method[35]. Deletion strains were generated by fusion PCR in *C. parapsilosis* CPL2H1, a *leu2/his1* double auxotrophic strain derived from *C. parapsilosis* CLIB214[19]. The oligonucleotides used to synthesise the deletion cassettes and to confirm the integrations are listed in Supplementary Data 1. Additional strains were constructed by CRISPR-Cas9 editing in *C. parapsilosis* CLIB214 using the pCP-tRNA plasmid ([35], Addgene # 133812). Suitable guides were computationally designed with EuPaGDT[72] to induce Cas9 cleavage within the first 25% of the targeted ORFs when no downstream in-frame methionine residues were present. When this was not possible, target sites were designed from the first 35%, 45% or 55% of the gene as necessary. Each guide RNA was generated by annealing two 23-bp oligonucleotides carrying appropriate overhanging ends (gRNA_TOP and gRNA_BOT, Supplementary Data 1). The short dsDNA was then cloned into the SapI-digested pCP-tRNA plasmid as described in ref. 35,73. The presence of the guide in the receiving plasmid was confirmed by PCR (M13FWD + relevant gRNA_BOT). Repair templates (RTs) were designed to repair the Cas9-induced double-strand break (DSB) by homology-directed repair (HDR), containing 30 bp homology arms to either side of the cut, 11 bp introducing stop codons in all reading frames, and a unique 20 bp barcode (tag). Each RT was generated by primer extension of overlapping oligonucleotides (Supplementary Data 1). For 22 mutants that were generated at an early stage of this study, the gRNAs and repair templates were designed manually (Supplementary Data 1, CLIB214 edited mutants, rows 4–25). All strains and primers are listed in Supplementary Data 1. All oligonucleotides were ordered from Eurofins Genomics. *C. parapsilosis* was transformed and screened as described in ref. 35,73. The regions around the edited sites were amplified by PCR and sequenced in all the strains from lineage A: if the desired edit was not present, lineage B was sequenced (Supplementary Data 1). Seven strains were excluded from the final library based on sequencing results. Three target genes were correctly edited only in lineage B (Supplementary Data 1).

Additional genes were deleted or edited in *C. parapsilosis* CLIB214 using CRISPR-Cas9 to follow up on phenotypes observed in the phenotypic screen of the library. These include *CPAR2_402070* (*MET4*), *CPAR2_702260* (*MET28*), *CPAR2_800700* (*PRK1*), *CPAR2_400120* (*SUL2*), *CPAR2_203100* (*SEO1*), *CPAR2_203280*, and *CPAR2_212620* (*AGP3*). The oligonucleotides were designed as described above (Supplementary Data 1, Follow-up CRISPR mutants). *MET28*, *PRK1*, *SUL2*, *SEO1* and *CPAR2_203280*, were deleted by replacing the entire *ORF* with the relevant barcodes, using longer (1–1.5 kb) RTs generated by fusion PCR (strains *met28Δ*, *prk1Δ*, *sul2Δ*, *seo1Δ*, and *203280Δ*, Supplementary Data 1).

The same strategy was used to delete *MET28* in the *met4Δ* background to create the *met4Δ/met28Δ* double mutant. The strains *Ed met4*, *Ed met28*, *Ed met4/met28*, *Ed sul2*, *Ed seo1*, *Ed 203280*, and *Ed agp3*, in which the genes were disrupted by the insertion of a barcode and premature stop codon, were obtained by transformation with the relevant plasmids and short dsDNA RTs (as described in the previous paragraph) (Supplementary Data 1). The changes introduced in the edited strains were complemented to create *cMET4*, *cPRK1*, and *cMET28*. The generation of *cMET28* required using a longer RT (made by fusion PCR), which restored the wildtype sequence but introduced 3 synonymous SNPs to prevent Cas9 from continuing to cut (Supplementary Data 1). *MET4* and *MET28* were disrupted using CRISPR-Cas9 in three additional *C. parapsilosis* strains: 02-203, 81-042, and CDC179 (Supplementary Data 1). The mutated loci were sequenced by Sanger sequencing (Supplementary Data 1).

CRISPR-Cas9 editing was used to insert a premature stop codon in *MET28* (*CTRG1_05070*) in *C. tropicalis* CAS08-102[35], as described in ref. 73. A suitable guide was cloned into pCT-tRNA ([35], Addgene # 133813), and an RT carrying the desired mutation was synthesised by primer extension and purified (Supplementary Data 1, Follow-up CRISPR mutants). The plasmid and the RT were electroporated into *C. tropicalis* CAS08-102 as described in ref. 73.

## Phenotype screening of the library

*C. parapsilosis* strains were pinned onto YPD agar plates using a 96-pin bolt replicator and used to inoculate 100 μL YPD media in 96-well microtiter plates. The strains were grown at 30 °C with gentle shaking to stationary phase, and then combined in a 1536 format array onto YPD agar plates (referred to as source plates) using a Singer Instruments ROTOR HDA. Independent disruptions (lineages A and B) were spotted in duplicate, providing a biological and technical replicate for each mutation. Where lineage B was not available, lineage A was spotted twice. The source plates were incubated at 30 °C for 24 h and then replicated to the different conditions (Supplementary Data 2, 3) using the Singer ROTOR HDA. Plates were imaged after 24 or 48 h (Supplementary Data 2, 3) with the Singer Instruments Phenobooth.

## Data analysis

Growth of the library on the phenotyping plates was assessed by measuring the median colony size for each strain (two biological and two technical replicates) using the open-source Phenobooth web-based application (https://singerinstruments.shinyapps.io/phenobooth/). Supplementary Data 1 shows the layout of the 1536 format array (1536 format plate tab) and the position in the 96-well stock plates (Plate layout tab).

Firstly, the growth of the library on the three media used as the base media for the phenotyping (YPD, SC, and YNB) was determined to identify strains that had a growth defect (Supplementary Data 2). Since, in this setting, we were comparing the growth of the strain within the same plate, the raw colony size was normalised based on row and column to control growth artifacts due to the position effect. The median colony size for the four replicates (two technical and two biological) of each mutant was calculated.

To compare the growth of the mutant strains to the control strains, we defined: (i) $Growth_{(CTRL)}$ as the mean of the normalised observations for CLIB214 and CPRI[19] (Mean WT, Supplementary Data 2); (ii) Growth Ratio relative to Mean WT as (median of the normalised size of mutant colony/$Growth_{(CTRL)}$) (Strain/WT RATIO in the Plate_YNB, Plate_YPD, and Plate_SC tabs, Supplementary Data 2).

Secondly, the growth of the library was evaluated on the different phenotyping media (Supplementary Data 3). In these experiments, the growth of each strain was compared across all the different conditions, so we used in the analysis the median of the raw colony size (i.e., not normalised), as artifacts due to the position of the strain on the plate would not be relevant. Median values were calculated from the raw colony size. To compare the growth of each strain on a test plate with the growth of the same strain on the equivalent base control condition, the natural Log Ratio (LogR in Supplementary Data 3: ln(median raw colony size test plate/median raw colony size base plate) was calculated. For each experimental plate, the mean and standard deviation of the distribution of the Log Ratio were calculated. Strains with a Z-score > 2 or < − 2 (Ln Ratio values that were greater than two standard deviations below (−) or above (+) the mean for each plate) were considered to have a growth defect (or advantage). These outlier strains are listed in Supplementary Data 3 (Tabs Outliers_YNBbased, Outliers_YPDbased, Outliers_SCbased). In the heat maps in Fig. 1, Supplementary Fig 3 and Supplementary Fig. 4, the outliers were colour-coded as follows: (i) strains with Z-scores between -3 and- 6 are indicated in yellow; (ii) strains with Z-score < −6 are indicated in orange; (iii) complete absence of growth is indicated in black. Data points for which at least one of the following conditions occurred were excluded from the analysis: (i) pinning issues; (ii) different behaviour of the two

independent lineages; (iii) contamination on plate; (iv) no growth on the CTRL plate (median growth less than 2).

## RNA-seq analysis

*C. parapsilosis* CLIB214 and mutant strains lacking *MET4* (*met4Δ*, 1 lineage), *MET28* (*met28Δ*, two lineages) or both genes (*met4Δ/met28Δ*, 1 lineage) were grown overnight in YPD media with shaking at 30 °C. Cells were washed twice in PBS and inoculated in 20 mL of SC + AA in duplicate to a final $A_{600}$ of 0.2. The cultures were grown at 30 °C with shaking to $A_{600}$ 1–1.5, centrifuged and resuspended in 20 mL of SC + AA or SC -cysteine/methionine, and then incubated for a further 2 h. The RNA extraction was performed following the protocol published by Cravener and Mitchell[74].

Strand-specific RNA-seq library preparation and sequencing were carried out by BGI. Paired-end reads (DNBSEQ, 2 × 100 bp) were obtained from three biological replicates from wild type (*Candida parapsilosis* CLIB214), *met4Δ*, and *met4Δ/met28Δ* incubated in the presence or absence of cysteine/methionine, while for *met28Δ* two replicates were obtained from one lineage and the third from the other lineage.

Samples were aligned to the genome[75] using STAR[76]. HTSeq[77] was used to count mapped reads per gene. Differentially expressed genes were identified using DESeq2[78]. Default parameters in DESeq2 were used.

Supplementary Data 4 lists the genes that were differentially expressed in the absence of cysteine and methionine in at least one of the strains tested (i.e., an adjusted *p*-value threshold of 0.05 and a $\log_2$ fold change threshold of −1 and 1).

The columns indicate the log fold change (logFC, highlighted in yellow if above the threshold), the adjusted *p*-value (P-val, highlighted in green if significant), the cumulative raw reads obtained in SC + AA (readA) or SC -cys/met (readB) for each strain (WT = CLIB214; MUT1 = *met4Δ*; MUT2 = *met28Δ*; DUAL = *met4Δ/met28Δ*). *C. albicans* and *S. cerevisiae* orthologs were obtained from the Candida Gene Order Browser (CGOB)[53].

## MEME-suite analysis

The MEME analysis was performed using 1-kb sequences upstream of the 45 sulphur-related genes induced in sulphur starvation (in red in Supplementary Table 2)[63]. MEME was used to identify motifs between 6–20 bp long and minimum 3 occurrences (reverse complement allowed) present in any number of sites per sequence. The statistically significant motifs were then used to interrogate the same input sequences using MAST, and hits with an *E*-value less than 10 were considered valid, which corresponded to a position *p*-value less than 0.0001[63].

## Statistical analysis

In the differential expression analysis, the Wald Test was used as implemented in the DESeq2 package. An implementation of the Benjamini-Hochberg false discovery rate (FDR) multiple test correction was applied in DESeq2. For biofilm biomass comparisons, statistical significance was calculated using the Kruskal Wallis test followed by post-hoc Dunn's test. The growth profiles of different strains using different sulphur sources were compared with the One-Way ANOVA and Tukey HSD test ($F = 222$, degrees of freedom = 29).

## Reporting summary

Further information on research design is available in the Nature Portfolio Reporting Summary linked to this article.

## Data availability

All data generated or analysed during this study are included in this published article, in Supplementary Data 1–4, and in the Supplementary Information file. All RNA-seq data is publicly available from SRA with BioProject accession number PRJNA1044317. Source data are provided in this paper.

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

## Acknowledgements
This work was supported by Science Foundation Ireland (19/FFP/6668, awarded to G.B.). Thanks to all the members of the Butler and Wolfe group for the comments on the manuscript.

## Author contributions
L.L. and G.B. conceived the concept and designed the experiments. L.L., L.I.S., E.O.C., F.M. and S.A.T. generated the collection of mutant strains. L.L., L.I.S. and E.O.C. performed the phenotypic tests. L.L. extracted the RNA for transcriptional analysis. C.O.B. and K.P.B. performed bioinformatics analyses. L.L. and G.B. wrote the manuscript. All authors commented on the manuscript. G.B. supervised the project.

## Competing interests
The authors declare no competing interests.
