## [Transparent Peer Review file · Nature Communications]

Alternative sulphur metabolism in the fungal pathogen *Candida parapsilosis*

Corresponding Author: Dr Lisa Lombardi

Version 0:

Reviewer comments:

Reviewer #1

(Remarks to the Author)

Lombardi et al., in the current manuscript have identified genes involved in the assimilation of inorganic and organic sulfur in *C. parapsilosis*. The authors have generated a collection of mutants in the fungus and screened this mutant collection on various stressors. They have demonstrated the role of MET4, MET 28 and MET 32 in utilization of sulfur in the fungus, and have elaborated on the genes that are regulated by these proteins. The manuscript is well written and results presented aptly. Despite the interesting observations vis a vis the diversification in roles of these genes in *C. parapsilosis* versus other yeasts, the study makes a limited advance in the current form.

Major concerns:

1. Although the authors have extensively described the methodology used for generating the mutant collection and have also explained the various phenotypes obtained with this mutant collection; they have not described those precise stressors that facilitated the identification of the MET genes!
2. The first section of the Results where the authors describe the phenotypic effects of disruptions is not required. Anyways this data is presented as Supplementary data. Instead the authors should list compounds that were used for screening mutants defective in sulfur assimilation or met/cys biosynthesis, which is more relevant for this study. How did the authors screen for met4 mutant as it grew well in the absence of methionine and cys?
3. In my opinion, the authors should explain the rationale for monitoring growth for 8 days?
4. Lines 609-613 in the Discussion; since the double delete of met4met28 exhibited abrogated growth on SC-M, doesn't the possibility of Met4 contributing in absence of Met28 be considered for answering the question posed by the authors? The statement sounds confusing and should be elaborated.
5. It would have been interesting to see if the met mutants affect the morphology of *C. parapsilosis* at all? or if their absence leads to a biofilm defect etc? making the study more clinically relevant.

Reviewer #2

(Remarks to the Author)

The current manuscript provides an excellent assessment of the methionine metabolism regulation of a fungal pathogen termed *Candida parapsilosis*. This organism is a relevant but not WHO critical fungal pathogen – the importance of the study would not be so much the key significance of *C. parapsilosis* as a cause of disease but the general interest of the evolutionary structuring of the metabolic circuitry.

The initial section describes the expansion of a collection of bar-coded null mutant strains in *C. parapsilosis*. This is certainly a very useful resource, but the collection itself is not centrally used in the analyses performed in this manuscript. Thus, this section could be removed, and the paper started with the simple observation that in *C. parapsilosis* a met4 mutant was not a methionine auxotroph.

The next section establishes that the met4 met28 double mutant is a met auxotroph, and that either Met4 or Met28 function is

sufficient to generate a prototroph. This result is general for *C. parapsilosis*, and not clade dependant. This adds to a complex story about Met4/Met28 in fungal methionine metabolism; in *S. cerevisiae* Met4 is central and Met28 supportive, in *C. albicans* Met4 is central and Met28 provides an unknown essential function, in filamentous fungi the Met4 ortholog is central and there is no Met28, in *S. pombe* the Met4 ortholog doesn't regulate methionine, and later in this paper we have an apparent essential Met4 with Met28 controlling methionine biosynthesis in *C. tropicalis*.

Given that both Met4 and Met28 are involved in the methionine circuitry of *C. parapsilosis*, the next data set investigates the specific roles of each TF. RNA seq establishes that Met28 is critical for regulating the primary genes of the standard methionine production pathways, while Met4 is critical for the expression of genes implicated in alternative sulphur metabolism. There are a small number of genes that are apparently regulated by both TFs; these include SUL2, AGP3, SEO1 – so does the knock-out of any of these genes create a methionine auxotroph?

Further analysis investigates in detail the role of the met circuitry control elements in sulphur metabolism. Met4 is found to be critical for growth in MOPS, SDS or DMSO as a sole source of sulphur for the cell, and Met32 was found to be equally important, a prediction generated from the expression profiling data. Met4 and Met32 are needed for use of organic, but not inorganic sulphur sources.

A major unaddressed question in the attempt to provide a comprehensive picture of the evolutionary pattern of fungal methionine metabolism are the regulators that are essential in some species, such as Met28 in *C. albicans*, Cbf1 in *C. parapsilosis*, and Met4 in *C. tropicalis*. This transfer of elements of the generic circuit to essentiality is fascinating, and while the current manuscript adds to the elements that have shifted to essentiality, it does not provide any framework for understanding these shifts.

Overall then, this is a really excellent paper. The experiments are very well done, the interpretations convincing, and the paper is very well written. A very comprehensive picture is provided of the methionine/sulphur metabolic circuitry in *C. parapsilosis*. However, in the end the paper provides basically an impressive assessment of the sulfur amino acid regulation in a minor fungal pathogen. The relationships of the circuit in *C. parapsilosis* to the circuits in other fungi are discussed, but we are really not provided fundamental insights into how and why these circuits differ in the different fungi, and I would expect such insights from a paper in a high profile generalist journal.

Minor point

References in the figure legends are provided as (first author, year) rather than a numeral.

Reviewer #3

(Remarks to the Author)

In this work the authors describe the rewiring of the regulatory circuit that controls the assimilation of sulphate and the synthesis of cysteine and methionine in *C. parapsilosis*, an important opportunistic human pathogen. By generating and screening a gene-deletion collection they found that unlike in other fungi, the transcription regulator Met4 is not essential for methionine synthesis in *C. parapsilosis*. Only when this regulator is disrupted together with its paralog Met28, cells become auxotrophic for methionine. Despite the functional redundancy, transcriptional profiling and further growth assays in a variety of sulphur sources revealed that Met28 has taken over the regulation of the methionine synthesis pathway, while Met4, instead, regulates alternative pathways for the assimilation of organosulfur compounds.

The article describes a very interesting case of transcriptional evolution, showing diversity in sulphur metabolism among closely related *Candida* species. The work is well executed, and the findings have implications for our understanding on how these human pathogens are able to thrive in the human host. My comments are minor, mainly trying to improve the way the data is presented.

1. After showing that either Met4 or Met28 are sufficient to grow in the absence of methionine, the question that naturally arises is how the functional redundancy is occurring at a molecular level. Subsequent transcriptional profiling seems directed to answer this question. However, in the abstract and in the results the authors do not follow this line of thought. Instead, they mainly centered on the divergence of the circuit being controlled by the two regulators, and only almost at the end of the discussion do they revisit the question. In my opinion the fact that the regulators are functionally redundant, but most of their targets have diverged is one of the most interesting aspects of the rewiring and should be put forward in the abstract and results.

2. The work devoted to the generation and screen of the deletion collection is considerable. However, currently, most of the results of the screen are presented as supplementary data and the first two sections of the results that refer to this work are quite superficial. Therefore, these results are sidelined and to some extent distractive from the main point of the article. I would suggest two options to improve this point: a) move out of the article most of the results of the screen and present them in a separate future manuscript, or b) strengthen the first two result sections and Figure 1. This could be done, for example, by replacing panels A to E of Figure 1 with the actual results of the screen presented in Supplementary Figure 1 (current panels A to D have anyway already been presented in references 35 and 52). The general results of the screen (how many mutants have a phenotype, overlap between media, phenotype conservation, etc.) could be also include in the two first results sections.

3. In Figure 4, in the volcano plots, it is not possible to see the main conclusion that wants to be drawn from it, that the deletion of Met4 and Met28 affects a different set of genes. A simple Venn diagram, some kind of correlation plot or marking the genes in the volcano plots as in Supplementary Figure 6 could help.
4. Not much is mentioned about the genes that are differentially expressed in both the met4 and met28 mutants. Could these explain the functional overlap? Can some of the genes not previously associated with sulphur metabolism but that change their expression in both mutants explain the functional redundancy of these regulators?
5. Nothing is mentioned about the cis-regulatory regions of the genes that change their expression under sulphur deprivation or specifically in the met4 and met28 mutants. Is there a common DNA motive that these regulators are binding? Can the differences in the cis elements explain the dissimilarities in the sets of genes controlled by these factors?, or is it rather coding changes in the bZIP domains of the regulators?
6. Is the identity of the bZIP domains of Met4 and Met28 as high in other species as in *C. parapsilosis*. Could this be informative in terms of the functional overlap?
7. What was the phenotype of the Met28 mutant in the large-scale screen of the library? Was it also unexpected?
8. The last sentence of the abstract could be phrased more carefully since there is no direct evidence to support the statement.
9. In the sentence starting on line 330, it would be informative to specify whether MET28 was differentially expressed.
10. In Figure 4, what does it mean that one half of the MET4 gene is marked in white? Also, does the color of the transmembrane transporters mean something and are they coded by the genes below the squares?
11. In the box plots of Figure 6, what do the boxes mean, standard deviation?
12. In line 305 it is mentioned that the phenotype of the Met28 in *C. tropicalis* is different from that of the same mutant in *C. albicans* and *C. parapsilosis*. However, the phenotype of the *C. albicans* mutant is not mentioned until the discussion.
13. In the legend of Supplementary Figure 5 it should be clearly stated that the difference with Figure 3 is the aligner that was used.
14. The following points could be modified to improve clarity:
 - Line 31, ... kill CLOSE TO/AROUND 1.5 million.
 - Line 32, beyond being eukaryotes, fungi are the closest organisms to animals.
 - Line 84, ...and then 3'-phospho-5'-..., "to" is missing.
 - Line 91, is it meant budding yeast?
 - Line 133, it is not clear what the prefixes Cd and Cm mean.
 - Line 177, replace "is" with "in".
 - Line 178, there is redundancy since histidine is an amino acid.
 - Details given in lines 188-189 seem irrelevant as this is only an example of an interesting mutant identified in the screen.
 - Lines 196 and 486, clearer if serial dilution ASSAYS would be used. Also, in the methods the term Spot assay is used. It would be easier to follow if a consistent term was employed.
 - Lines 223 to 229, the details of the structure of Met4 seem irrelevant, especially because they are latter given in the legend of Figure 3.
 - The genus of *Ogataea parapolyomorpha* should be spelled only the first time the species is mentioned and from there on only use *O.* the same with *Aspergillus*.
 - At the top of Figure 3, there is a legend that does not seem to be needed: PhyML...
 - Line 323, "and" can be deleted.
 - Line 530, do you mean ScMet4?
 - Line 540, Cbf1 seem more appropriate than the gene name.

- Check syntax of the end of sentence in lines 552-554

- In Figure 7, what are the white boxes in the promoters? Also in this figures, why is a single node leading to *C. par.*, *C. tro* and *C. alb*?

- In line 685 it is not necessary to describe the composition of the SC medium as it was defined in the previous paragraph.

- The files of the supplementary tables are missing a clear title to be identified as Supplementary Table 1, 2, etc.

Version 1:

Reviewer comments:

Reviewer #1

(Remarks to the Author)

The authors have significantly improved the manuscript, and all comments have been addressed satisfactorily. I appreciate the authors efforts to perform the biofilm experiments. Given the description of these results in the main text, it would be easy for the reader to have the figure too as part of the main text instead of being placed as supplementary figure.

Reviewer #2

(Remarks to the Author)

The modifications that were done to the manuscript are extensive and have certainly improved the paper considerably. I have a couple of minor comments, but feel the manuscript in its current form is suitable for publication.

Comb jellies are now considered the closest relatives to animals, Nature 618, pages 110–117 (2023) so the statement about fungi and animals can be rephrased.

Line 77 - Is it only sulphur metabolism that is poorly understood? How about phosphorus, nitrogen... How do you quantify the relative understanding? I would reword this statement.

Line 341 By contrast,

Reviewer #3

(Remarks to the Author)

All my initial comments were addressed in the revised version of the manuscript, improving it considerably. Below are a couple of minor suggestions that could be incorporated for the final version of the article.

1. Could something more general be said about the overall conservation of function/phenotype of the genes that were characterized in the screen. Many concrete examples of functional conservation/divergence with *C. albicans* are included in the first sections of the results, but not an overall trend. For example, are phenotypes overall more conserved with *C. albicans* or other CTG species than what has been observed between *C. albicans* and *S. cerevisiae*?

2. In my opinion Figure 8 is quite difficult to understand because it has many unnecessary details. For example, the names of DNA binding domains (bZIP, etc.) do not seem necessary, as well as the INT and AD labels. The RYAATNN binding motif also seems dispensable and even misleading in *C. parapsilosis* as Met28 is not binding it in this species. I would also increase the space between Met28 and the DNA in *C. parapsilosis* so that it is clear that it is not binding, especially in class 2 genes.

3. Have the authors generated the met32/met28 double mutant? If this mutant is auxotrophic to methionine it would strengthen the role of MET32 in the pathway and its possible association with Met4.

4. When the enriched motifs are described I do not think it is necessary to give the details of the repetitive motifs, just mention them without providing the motifs themselves.

5. Is the term "transcriptional footprints" correctly employed? If I am correct, the footprint of a transcription factor refers to the area/region where the TF is binding DNA, but I am not sure this is what is meant here.

6. Sentence in line 155 that refers to mutants previously generated could state that Holland refers to strains made by the authors of the current article. This is important so that it is clear that they are the same strains mentioned in sentence starting in line 144.

7. In the first two sections of the results there are several short paragraphs that in my opinion belong together in larger paragraphs.
8. End of line 194 there is an extra parenthesis at the end of the sentence.
9. Line 434 should say “Met4” instead of “Met”.
10. Line 170, substitute “N” for “nitrogen” as is used in the rest of the manuscript.
11. Slashes in the abstract (lines 23 and 25) could be substituted by “and”.

We would like to thank the reviewers for their comments on the manuscript. We have been conducting additional work to properly address the concerns and comments raised, and the results significantly reshaped our interpretation of the regulatory network controlling sulphur metabolism in *C. parapsilosis*.

We now suggest that in *C. parapsilosis*, as in *S. cerevisiae*, Met4 is recruited to promoters by its interaction with Met32. However, in *C. parapsilosis* Met28 – and not Met4 – is the core regulator of cysteine and methionine synthesis most likely due to its interaction with the DNA-binding protein Cbf1. We hypothesise that Met4 can contribute to assimilation of inorganic sulphur in the absence of Met28, which would explain the functional redundancy. Due to differences in the regulation of *MET4*, *MET28*, *CBF1* and *MET32*, the two branches of sulphur metabolism (assimilation of inorganic sulphur and synthesis of methionine on one side, import and assimilation of organic sulphur sources on the other) appear to be relatively less intertwined in *C. parapsilosis* compared to *S. cerevisiae*, which is also an additional prerequisite for functional redundancy between Met4 and Met28.

Overall, we are confident that our manuscript was significantly improved by addressing the reviewers comments.

Detailed answers to the comments may be found below.

REVIEWER COMMENTS

Reviewer #1 (Remarks to the Author):

Lombardi et al., in the current manuscript have identified genes involved in the assimilation of inorganic and organic sulfur in *C. parapsilosis*. The authors have generated a collection of mutants in the fungus and screened this mutant collection on various stressors. They have demonstrated the role of *MET4*, *MET 28* and *MET 32* in utilization of sulfur in the fungus, and have elaborated on the genes that are regulated by these proteins. The manuscript is well written and results presented aptly. Despite the interesting observations vis a vis the diversification in roles of these genes in *C. parapsilosis* versus other yeasts, the study makes a limited advance in the current form.

Major concerns:

1. Although the authors have extensively described the methodology used for generating the mutant collection and have also explained the various phenotypes obtained with this mutant collection; they have not described those precise stressors that facilitated the identification of the *MET* genes!

Thank you for your comment. In the revised manuscript, the screen that led to the identification of *MET4* is described in lines 233-238, and more in detail in Supplementary Note 2 and Supplementary Figure 4. The ability of the strains in the library to grow in the absence of specific classes of amino acids was tested as described in methods (lines 695-702); the amino acids missing in each condition tested are listed above the heatmap in Supplementary Figure 4.

2. The first section of the Results where the authors describe the phenotypic effects of disruptions is not required. Anyways this data is presented as Supplementary data. Instead the authors should list compounds that were used for screening mutants defective in sulfur assimilation or met/cys biosynthesis, which is more relevant for this study. How did the authors screen for *met4* mutant as it grew well in the absence of methionine and cys?

The library was screened for the ability to grow in the absence of different classes of amino acids (Supplementary Figure 4); two of these classes were lacking cysteine and methionine. We therefore anticipated that the *met4Δ* mutant would fail to grow. However, this was not the case. This lack of a phenotype, in a sense, was what prompted us to further investigate the

role of *MET4* and its paralog *MET28* in the synthesis of cysteine and methionine. In the spot assays performed as follow up to the screen depicted in Supplementary Figure 4, we tested the ability of selected mutants to grow in the absence of cysteine, methionine, or both (Fig 3). We could only see a growth defect when both *MET4* and *MET28* were absent, which explained why we did not see a phenotype for the individual mutants in the library screen. We changed the manuscript to hopefully make this clearer for the reader (lines 233-238). The media used for the spot assay is described in Fig. 3 legend and more in detail in Methods (lines 704-707). The media used for screening defects in sulphur assimilation (Fig 7) is described in lines 494-499, and more in detail in Methods (lines 713-731).

3. In my opinion, the authors should explain the rationale for monitoring growth for 8 days? Thank you for this suggestion. The rationale is now indicated in lines 728-730: "Although 6 days of incubation was sufficient for robust growth of *L. elongisporus* CBS2605⁶², the OD600 was measured after 8 days of incubation to facilitate the slower growth of *C. parapsilosis* CLIB214".

4. Lines 609-613 in the Discussion; since the double delete of *met4met28* exhibited abrogated growth on SC-M, doesn't the possibility of Met4 contributing in absence of Met28 be considered for answering the question posed by the authors? The statement sounds confusing and should be elaborated.

In light of the new data acquired at the revision stage, the Discussion was extensively modified. The possibility of Met4 contributing in the absence of Met28 is now part of the model that we propose for sulphur assimilation in *C. parapsilosis*, as stated in lines 624-628, and described in Fig. 8.

5. It would have been interesting to see if the *met* mutants affect the morphology of *C. parapsilosis* at all? or if their absence leads to a biofilm defect etc? making the study more clinically relevant.

We agree with the reviewer that this would increase the significance of this work. We did test biofilm formation in strains in which *MET4*, *MET28*, or both were disrupted, in the presence and absence of methionine. The results are described in a new paragraph (lines 518-535) and in Supplementary Figure 8/Supplementary Table 9. In the presence of methionine, loss of both *MET4* and *MET28*, but not either individually, significantly reduces biofilm formation. In the absence of methionine, the wild-type strain and cells lacking *MET4* produce comparable biofilms, though overall biofilm is reduced compared to when methionine is present. However, loss of *MET28* alone completely abolishes biofilm formation, reducing it to the level of the double disrupted strain. Our results suggest that Met28 is a core regulator of biofilm formation during sulphur limitation, suggesting that it is an important virulence factor.

Reviewer #2 (Remarks to the Author):

The current manuscript provides an excellent assessment of the methionine metabolism regulation of a fungal pathogen termed *Candida parapsilosis*. This organism is a relevant but not WHO critical fungal pathogen – the importance of the study would not be so much the key significance of *C. parapsilosis* as a cause of disease but the general interest of the evolutionary structuring of the metabolic circuitry.

The initial section describes the expansion of a collection of bar-coded null mutant strains in *C. parapsilosis*. This is certainly a very useful resource, but the collection itself is not centrally used in the analyses performed in this manuscript. Thus, this section could be removed, and the paper started with the simple observation that in *C. parapsilosis* a *met4* mutant was not a methionine auxotroph.

Based on the comments from all the reviewers and the recommendations of the editor, we decided to include the description of the library in the manuscript. However, we do agree with Reviewer #2 that the way in which the manuscript was organised did not put the

generation of the library in the correct perspective. We therefore changed the structure of the Results to focus on the findings that we obtained screening the library (in terms of functional conservation with *C. albicans* and *S. cerevisiae* and previously unidentified roles of transcription factors and protein kinases), rather than on the generation of the library. We now describe these in the first two paragraphs (lines 155-171, and 175-229) of the Results and Fig. 1.

The next section establishes that the *met4 met28* double mutant is a met auxotroph, and that either Met4 or Met28 function is sufficient to generate a prototroph. This result is general for *C. parapsilosis*, and not clade dependant. This adds to a complex story about Met4/Met28 in fungal methionine metabolism; in *S. cerevisiae* Met4 is central and Met28 supportive, in *C. albicans* Met4 is central and Met28 provides an unknown essential function, in filamentous fungi the Met4 ortholog is central and there is no Met28, in *S. pombe* the Met4 ortholog doesn't regulate methionine, and later in this paper we have an apparent essential Met4 with Met28 controlling methionine biosynthesis in *C. tropicalis*.

Given that both Met4 and Met28 are involved in the methionine circuitry of *C. parapsilosis*, the next data set investigates the specific roles of each TF. RNA seq establishes that Met28 is critical for regulating the primary genes of the standard methionine production pathways, while Met4 is critical for the expression of genes implicated in alternative sulphur metabolism. There are a small number of genes that are apparently regulated by both TFs; these include *SUL2*, *AGP3*, *SEO1* – so does the knock-out of any of these genes create a methionine auxotroph?

Thank you for raising this important question. We addressed it by generating individual mutants of *SUL2*, *AGP3*, *SEO1*, and *CPAR2_203280* (which we included because of its 88% percentage of identity with *SEO1*) in *C. parapsilosis* CLIB214, and testing their ability to synthesize cysteine and methionine. The results are described in lines 421-426 and Supplementary Figure 7. We find that disrupting each gene individually does not result in methionine auxotrophy. It is possible however that if all genes were simultaneously deleted methionine synthesis would be disrupted.

Further analysis investigates in detail the role of the met circuitry control elements in sulphur metabolism. Met4 is found to be critical for growth in MOPS, SDS or DMSO as a sole source of sulphur for the cell, and Met32 was found to be equally important, a prediction generated from the expression profiling data. Met4 and Met32 are needed for use of organic, but not inorganic sulphur sources.

A major unaddressed question in the attempt to provide a comprehensive picture of the evolutionary pattern of fungal methionine metabolism are the regulators that are essential in some species, such as Met28 in *C. albicans*, Cbf1 in *C. parapsilosis*, and Met4 in *C. tropicalis*. This transfer of elements of the generic circuit to essentiality is fascinating, and while the current manuscript adds to the elements that have shifted to essentiality, it does not provide any framework for understanding these shifts.

We have substantially changed the discussion, and added a new version of the model proposed (Fig.8). We suggest that the core circuit of 4 transcription factors (Cbf1, Met32, Met4 and Met28) and possibly also Met30 is conserved between *S. cerevisiae* and *Candida* (and probably other yeasts) but there are differences in the specific genes regulated and the detailed roles of the transcription factors. We now discuss the evolution of the circuit from a single transcription factor (Met4) to a complex. We believe we have now provided a more detailed framework.

Overall then, this is a really excellent paper. The experiments are very well done, the interpretations convincing, and the paper is very well written. A very comprehensive picture is provided of the methionine/sulphur metabolic circuitry in *C. parapsilosis*. However, in the end the paper provides basically an impressive assessment of the sulfur amino acid

regulation in a minor fungal pathogen. The relationships of the circuit in *C. parapsilosis* to the circuits in other fungi are discussed, but we are really not provided fundamental insights into how and why these circuits differ in the different fungi, and I would expect such insights from a paper in a high profile generalist journal.

As noted above, we now provide a more detailed description of the possible evolution of the sulphur circuit in yeasts. It is difficult to explain “why”, but we do discuss our observation that *C. parapsilosis* encodes many more organosulfur transporters than other yeasts (regulated by Met4) and that this may facilitate its growth in ecological niches, such as in the human host. We also note that regulation of the use of inorganic and organic sources of sulphur is separated in *C. parapsilosis*, unlike in *S. cerevisiae*. This again facilitates the use of different sulphur sources in different niches.

Minor point

References in the figure legends are provided as (first author, year) rather than a numeral. Thank you, we addressed this in all the legends.

Reviewer #3 (Remarks to the Author):

In this work the authors describe the rewiring of the regulatory circuit that controls the assimilation of sulphate and the synthesis of cysteine and methionine in *C. parapsilosis*, an important opportunistic human pathogen. By generating and screening a gene-deletion collection they found that unlike in other fungi, the transcription regulator Met4 is not essential for methionine synthesis in *C. parapsilosis*. Only when this regulator is disrupted together with its paralog Met28, cells become auxotrophic for methionine. Despite the functional redundancy, transcriptional profiling and further growth assays in a variety of sulphur sources revealed that Met28 has taken over the regulation of the methionine synthesis pathway, while Met4, instead, regulates alternative pathways for the assimilation of organosulfur compounds.

The article describes a very interesting case of transcriptional evolution, showing diversity in sulphur metabolism among closely related *Candida* species. The work is well executed, and the findings have implications for our understanding on how these human pathogens are able to thrive in the human host. My comments are minor, mainly trying to improve the way the data is presented.

1. After showing that either Met4 or Met28 are sufficient to grow in the absence of methionine, the question that naturally arises is how the functional redundancy is occurring at a molecular level. Subsequent transcriptional profiling seems directed to answer this question. However, in the abstract and in the results the authors do not follow this line of thought. Instead, they mainly centered on the divergence of the circuit being controlled by the two regulators, and only almost at the end of the discussion do they revisit the question. In my opinion the fact that the regulators are functionally redundant, but most of their targets have diverged is one of the most interesting aspects of the rewiring and should be put forward in the abstract and results.

Thank you for this comment. Due to the additional analyses performed at the revision stage, the manuscript went through extensive changes. The partial functional redundancy between Met4 and Met32 achieved is now a central theme of the paper, and it has been put forward in the Abstract (lines 21-30). We now suggest that in *C. parapsilosis*, as in *S. cerevisiae*, Met4 is recruited to promoters by its interaction with Met32. However, in *C. parapsilosis* Met28 – and not Met4 – is the core regulator of cysteine and methionine synthesis most likely due to its interaction with the DNA-binding protein Cbf1. We hypothesise that Met4 can contribute to assimilation of inorganic sulphur in the absence of Met28, which would explain the functional redundancy. Due to differences in the regulation of *MET4*, *MET28*, *CBF1* and *MET32*, the two branches of sulphur metabolism (assimilation of inorganic sulphur and

synthesis of methionine on one side, import and assimilation of organic sulphur sources on the other) appear to be relatively less intertwined in *C. parapsilosis* compared to *S. cerevisiae*, which is also an additional prerequisite for functional redundancy between Met4 and Met28. Our new model is described in the Discussion (lines 579-628) and in Fig. 8.

2. The work devoted to the generation and screen of the deletion collection is considerable. However, currently, most of the results of the screen are presented as supplementary data and the first two sections of the results that refer to this work are quite superficial. Therefore, these results are sidelined and to some extent distractive from the main point of the article. I would suggest two options to improve this point: a) move out of the article most of the results of the screen and present them in a separate future manuscript, or b) strengthen the first two result sections and Figure 1. This could be done, for example, by replacing panels A to E of Figure 1 with the actual results of the screen presented in Supplementary Figure 1 (current panels A to D have anyway already been presented in references 35 and 52). The general results of the screen (how many mutants have a phenotype, overlap between media, phenotype conservation, etc.) could be also include in the two first results sections.

Based on the comments of the editor, we decided to strengthen the first two Results sections and Figure 1. We did move panels A to E to Supplementary Figure 1. We did include the previous panel E in the new Fig 1 (as panel A), and expanded on the general results of the screen in the first paragraph (Lines 160-171). The second paragraph in the Results (lines 175-229) now focuses on the screen of the library in the presence of different stressors (cell wall, osmotic, antifungal drugs etc.) or different nitrogen sources (Fig 1B and C). We also comment on our findings in terms of functional conservation with *C. albicans* and *S. cerevisiae* and previously unidentified roles of transcription factors and protein kinases.

3. In Figure 4, in the volcano plots, it is not possible to see the main conclusion that wants to be drawn from it, that the deletion of Met4 and Met28 affects a different set of genes. A simple Venn diagram, some kind of correlation plot or marking the genes in the volcano plots as in Supplementary Figure 6 could help.

We really appreciate that this was pointed out, and we completely agree. We modified Fig 4 by introducing two circles surrounding the two different sets of genes that we refer to in the main text: the ones that are not induced in the absence of *MET4* and the ones that are not induced in the absence of *MET28*. We included next to the circles the functional categories of these genes (e.g. sulphate assimilation, Met metabolism etc.) as described in the main text and in Supplementary Table 6. We believe that the message that we want to convey is much clearer now.

4. Not much is mentioned about the genes that are differentially expressed in both the met4 and met28 mutants. Could these explain the functional overlap? Can some of the genes not previously associated with sulphur metabolism but that change their expression in both mutants explain the functional redundancy of these regulators?

Thank you for this comment. We addressed it by generating mutant strains lacking the genes that were regulated equally by Met4 and Met28 based on our transcriptional analysis (*SUL2*, *AGP3*, *SEO1*, and *CPAR2_203280*, which we included because of its 88% percentage of identity with *SEO1*) in *C. parapsilosis* CLIB214. We tested their ability to synthesize cysteine and methionine. The results are described in lines 421-426 and Supplementary Figure 7. We find that disrupting each gene individually does not result in methionine auxotrophy. It is possible however that if all genes were simultaneously deleted methionine synthesis would be disrupted.

5. Nothing is mentioned about the cis-regulatory regions of the genes that change their expression under sulphur deprivation or specifically in the met4 and met28 mutants. Is there a commo DNA motive that these regulators are binding? Can the differences in the cis elements explain the dissimilarities in the sets of genes controled by these factors?, or is it rather coding changes in the bZIP domains of the regulators?

We are grateful for this comment: in fact, addressing it significantly influenced the interpretation of our data. We used the MEME suite to find significant motifs (described in Methods, lines 896-903), expecting to find sites that Met4 and Met28 would recognize and bind. Instead, the promoters of sulphur-responsive genes in *C. parapsilosis* are enriched in binding motifs recognized by the DNA-binding proteins Cbf1 and Met32. The binding pattern suggested by the binding sites mapping overlaps with our transcriptional data in a way that suggests that what we originally identified as the transcriptional effect of two proteins (Met4 and Met28) is likely the result of the two complexes (Met4-Met32 and Met28-Cbf1) binding to the promoters (Fig 6). Indeed, differences in the *cis* elements explain the dissimilarities in the sets of genes controlled by Met4 and Met28. These results are described in lines 439-489, and put into perspective in a new model of sulphur metabolism in *C. parapsilosis* depicted in Fig 8. Parts of the discussion are also dedicated to this (lines 598-628).

6. Is the identity of the bZIP domains of Met4 and Met28 as high in other species as in *C. parapsilosis*. Could this be informative in terms of the functional overlap?

In light of the analysis performed on the *cis* elements (see point 5), we think it's unlikely that Met4 and Met28 directly bind DNA. Instead, we discuss the likely effects of interacting with Cbf1 and Met32.

7. What was the phenotype of the Met28 mutant in the large-scale screen of the library? Was it also unexpected?

The library contains a *met28Δ* mutant; however, this strain did not show a growth defect when tested in the absence of cysteine and methionine (and in fact it is not present among the mutants listed in Supplementary Figure 4). The spot assay in the absence of cysteine and methionine (Fig 3) confirmed this observation. This is different than what observed in *S. cerevisiae*, where deletion of *MET28* results in methionine auxotrophy.

8. The last sentence of the abstract could be phrased more carefully since there is no direct evidence to support the statement.

We agree. The sentence was removed from the abstract.

9. In the sentence starting on line 330, it would be informative to specify whether MET28 was differentially expressed.

We added the sentence "On the contrary, the transcriptional level of *MET28* is not affected by cysteine/methionine levels" (lines 341-342). This is different than what observed in *S. cerevisiae*, as discussed in lines 613-628).

10. In Figure 4, what does it mean that one half of the MET4 gene is marked in white? Also, does the color of the transmembrane transporters mean something and are they coded by the genes below the squares?

Thank you for pointing this out. We added two sentences to the legend of Fig 5 to make this clearer. "We could not determine if *MET4* regulates itself because our data was obtained in a *met4Δ* strain: for this reason, half of the rectangle was left white." The second sentence reads: "The boxes representing the transporters are colour-coded based on their putative specificity for sulphate (teal), organosulfur compounds (green and yellow), or amino acids (purple)".

11. In the box plots of Figure 6, what do the boxes mean, standard deviation?

The legend was modified as follows: "Individual observations are plotted, and the median is indicated by the horizontal bars. The box represents the range of data. Detailed information on the number of independent replicates, mean, and standard deviation is available in Supplementary Table 8. Statistical analysis One Way ANOVA. ns = not significant, **** P ≤ 0.0001. n = minimum 3".

12. In line 305 it is mentioned that the phenotype of the Met28 in *C. tropicalis* is different

from that of the same mutant in *C. albicans* and *C. parapsilosis*. However, the phenotype of the *C. albicans* mutant is not mentioned until the discussion.

Thank you for this comment. We now mentioned the different phenotype of Met28 in *C. albicans* in the Introduction (lines 109-110).

13. In the legend of Supplementary Figure 5 it should be clearly stated that the difference with Figure 3 is the aligner that was used.

Thank you for this comment. We changed the figure legend to: "Protein sequences retrieved from YGOB²³, CGOB²⁴ or the indicated accession numbers were aligned using Muscle (**as opposed to ClustalO**) implemented in Seaview²⁵. Trees were inferred using PhyML restricted to conserved regions selected using Gblocks. WGD = Whole Genome Duplication; KLE = *Kluyveromyces/Lachancea/Emmentothecium*."

14. The following points could be modified to improve clarity:

- Line 31, ... kill CLOSE TO/AROUND 1.5 million.

We changed the sentence to "kill around 1.5 million.." (line 34).

- Line 32, beyond being eukaryotes, fungi are the closest organisms to animals.

This information was included in line 35.

- Line 84, ...and then 3'-phospho-5'-..., "to" is missing.

This was changed (line 87)

- Line 91, is it meant budding yeast?

In this case the sentence refers to yeasts in the Saccharomycotina for which genes involved in sulphur metabolism are known.

- Line 133, it is not clear what the prefixes Cd and Cm mean.

Candida dubliniensis and *Candida maltosa* is now spelled in extenso (lines 142-143).

- Line 177, replace "is" with "in".

Thank you, we fixed it (now line 163).

- Line 178, there is redundancy since histidine is an amino acid.

We corrected this throughout the manuscript.

- Details given in lines 188-189 seem irrelevant as this is only an example of an interesting mutant identified in the screen.

The results of the screen now are described more in depth in the first two paragraphs of the Results.

- Lines 196 and 486, clearer if serial dilution ASSAYS would be used. Also, in the methods the term Spot assay is used. It would be easier to follow if a consistent term was employed.

Due to text rearrangements, we erased line 196. We changed the text in line 486 and in the Methods.

- Lines 223 to 229, the details of the structure of Met4 seem irrelevant, especially because they are latter given in the legend of Figure 3.

The sentence was shortened (lines 252-255), and the details were integrated into the legend of Fig 2 (previous Fig 3).

- The genus of *Ogataea parapolyomorpha* should be spelled only the first time the species is mentioned and from there on only use *O.* the same with *Aspergillus*.

Thank you, we corrected this.

- At the top of Figure 3, there is a legend that does not seem to be needed: PhyML...
Thank you for pointing this out; we removed it.

- Line 323, "and" can be deleted.
Line 332: "and" was removed.

- Line 530, do you mean ScMet4?
The sentences was changed to make it clearer (line 555).

- Line 540, Cbf1 seem more appropriate than the gene name.
We changed this (line 565).

- Check syntaxis of the end of sentence in lines 552-554
The sentence was rephrased to: "We found that *C. parapsilosis* can use organosulfur compounds as sole sulphur sources as efficiently as it uses ammonium sulphate (a source of inorganic sulphur) (Fig 7B)" (lines 652-654).

- In Figure 7, what are the white boxes in the promoters? Also in this figures, why is a single node leading to *C. par.*, *C. tro* and *C. alb*?

Fig 8 (previously Fig 7) was changed extensively in light of the new results described in the revised manuscript. There are now two different nodes (one for *C. albicans* and *C. tropicalis*, and the other for *C. parapsilosis*).

- In line 685 it is not necessary to describe the composition of the SC medium as it was defined in the previous paragraph.

Lines 704-707: thank you, we corrected this.

- The files of the supplementary tables are missing a clear title to be identified as Supplementary Table 1, 2, etc.

We report below the titles of all the Supplementary Tables:

- Supplementary Table 1
- Supplementary Table 2. *C. parapsilosis* CLIB214 genes involved in different cellular processes based on phenotypic screening.
- Supplementary Table 3
- Supplementary Table 4
- Supplementary Table 5
- Supplementary Table 6. Upregulated genes of *C. parapsilosis* CLIB214 upon Cys/Met starvation and their expression in the *met4* and *met28* single and double deletion mutants.
- Supplementary Table 7. Binding motifs recognized by Met32 and Cbf1 in the promoters of sulphur-related genes induced under Cys/Met starvation.
- Supplementary Table 8. Use of organosulfur compounds as only sulphur source. AS: ammonium sulphate; SD: standard deviation; SEM: standard error of mean).
- Supplementary Table 9. Three independent experiments testing biofilm formation (dry weight, mg) in the presence and absence of Met. The different lineages tested are also indicated; NT: not tested; SD: standard deviation; SEM: standard error of mean.

REVIEWERS' COMMENTS

Reviewer #1 (Remarks to the Author):

The authors have significantly improved the manuscript, and all comments have been addressed satisfactorily.

I appreciate the authors' efforts to perform the biofilm experiments. Given the description of these results in the main text, it would be easy for the reader to have the figure too as part of the main text instead of being placed as supplementary figure.

We thank the reviewer for the positive feedback.

Figure 7 has been moved to the main text.

Reviewer #2 (Remarks to the Author):

The modifications that were done to the manuscript are extensive and have certainly improved the paper considerably. I have a couple of minor comments, but feel the manuscript in its current form is suitable for publication.

Comb jellies are now considered the closest relatives to animals, Nature 618, pages 110–117 (2023) so the statement about fungi and animals can be rephrased.

Thank you for pointing this out. The incorrect sentence has now been removed.

Line 77 - Is it only sulphur metabolism that is poorly understood? How about phosphorus, nitrogen... How do you quantify the relative understanding? I would reword this statement.

The previous statement has been rephrased as follows: "Sulphur metabolism is of particular interest, as it is not conserved between host and fungal cells, making it a promising drug target²⁴."

Line 341 change "On the contrary" to "By contrast" in line 341 of the manuscript "On the contrary" was replaced with "By contrast".

Reviewer #3 (Remarks to the Author):

All my initial comments were addressed in the revised version of the manuscript, improving it considerably. Below are a couple of minor suggestions that could be incorporated for the final version of the article.

1. Could something more general be said about the overall conservation of function/phenotype of the genes that were characterized in the screen. Many concrete examples of functional conservation/divergence with *C. albicans* are included in the first sections of the results, but not an overall trend. For example, are phenotypes overall more conserved with *C. albicans* or other CTG species than what has been observed between *C. albicans* and *S. cerevisiae*?

As noted, we have included some description of functional conservation/divergence. We considered attempting to do a more general comparison. However, we feel that this would make the manuscript very unwieldy. In addition, we do not have a complete set of gene disruptions in *C. parapsilosis*, which would bias the results. For these reasons we have not expanded the description.

2. In my opinion Figure 8 is quite difficult to understand because it has many unnecessary details. For example, the names of DNA binding domains (bZIP, etc.) do not seem necessary, as well as the INT and AD labels. The RYAATNN binding motif also seems dispensable and even misleading in *C. parapsilosis* as Met28 is not binding it in this species. I would also increase the space between Met28 and the DNA in *C. parapsilosis* so that it is clear that it is not binding, especially in class 2 genes.

We thank the reviewer for this comment: the RYAATNN binding motif should not have been included in the *C. parapsilosis* panel, as we have no data to support its presence in this species. It was an error. We also eliminated the labels indicating the domains of the different proteins, except the activation domain (AD). We left the Ad domain because it helps with visualizing an important difference between the two species shown in the working model: *C. parapsilosis* Met28, unlike *S. cerevisiae* Met28, seems to have the ability to activate transcription in the absence of Met4. The figure legend was also changed accordingly.

3. Have the authors generated the met32/met28 double mutant? If this mutant is auxotrophic to methionine it would strengthen the role of MET32 in the pathway and its possible association with Met4.

We haven't yet generated such a mutant, but we agree with the reviewer that it is a sensible next step. We do not believe that it is necessary for the current manuscript.

4. When the enriched motifs are described I do not think it is necessary to give the details of the repetitive motifs, just mention them without providing the motifs themselves.

Thanks for the comment. However, we think that it is somewhat clearer to include the actual repetitive sequences, rather than refer to information "not shown".

5. Is the term "transcriptional footprints" correctly employed? If I am correct, the footprint of a transcription factor refers to the area/region where the TF is binding DNA, but I am not sure this is what is meant here.

The sentence has been changed to: "Our findings suggest that the transcriptional effects of Met4 and Met28 are likely achieved through association with Met32 and Cbf1, respectively."

6. Sentence in line 155 that refers to mutants previously generated could state that Holland refers to strains made by the authors of the current article. This is important so that it is clear that they are the same strains mentioned in sentence starting in line 144.

This information was added. The sentence now reads: "Overall, 253 new mutant strains were generated. When combined with the strains previously generated in our laboratory by Holland et al.19, the collection includes disruptions of 200 predicted transcription factors, 85 predicted protein kinases, and 66 genes with miscellaneous functions, most of which were generated as two independent lineages (Fig 1A)."

7. In the first two sections of the results there are several short paragraphs that in my opinion belong together in larger paragraphs.

The paragraphs were joined as suggested.

8. End of line 194 there is an extra parenthesis at the end of the sentence.
Thank you, the extra parenthesis was removed.

9. Line 434 should say "Met4" instead of "Met".
Thank you for noticing, we changed Met to Met4.

10. Line 170, substitute "N" for "nitrogen" as is used in the rest of the manuscript.
We replaced N with nitrogen.

11. Slashes in the abstract (lines 23 and 25) could be substituted by "and".
We substitute the slashes with "and" in both cases.